

# Balloon Observations Suggesting Sea Salt Injection into the Stratosphere from Hunga Tonga-Hunga Ha'apai

Hazel Vernier[1,2], Demilson Quintão[3], Bruno Biazon[3], Eduardo Landulfo[4], Giovanni Souza[4], V. Amanda Santos[4], J. S. Fabio Lopes[4], C.P. Alex Mendes[4], A.S. José da Matta[4,5], K. Pinheiro Damaris[6], Benoit Grosslin[7], P.M. P. Maria Jorge[8], Maria de Fátima Andrade[8,9], Neeraj Rastogi[10], Akhil Raj[11], Hongyu Liu[12,13], Mahesh Kovilakam[12], Suvarna Fadnavis[14], Frank G.Wienhold[15], Mathieu Colombier[16], D. Chris Boone[17], Gwenael Berthet[2], Nicolas Dumelie[1], Lilian Joly[1], Jean-Paul Vernier[12,13,1]

[1]Groupe de Spectrométrie Moléculaire et Atmosphérique, Universitéde Reims, Champagne-Ardenne, 51687, France
[2]Laboratoire de de Physique et de Chimie de l'Environnement et de l'Espace, Orleans, 45071, France
[3]Instituto de Pesquisas Meteorológicas - IPMet – UNESP,Chácaras Bauruenses, Bauru - SP, 17048-699, Brazil
[4]Instituto de Pesquisas Energéticas e Nucleares, University of São Paulo Butantã, São Paulo - SP, 05508-000, Brazil
[5]Escola Politécnica da Universidade de São Paulo, Butantã, São Paulo - SP, 05508-010, Brazil
[6]Departamento de Engenharia Química UFSM, Camobi, Santa Maria - State of Rio Grande do Sul, 97105-340, Brazil
[7]Institut de Combustion Aérothermique Réactivité Environnement (ICARE), 45100 Orléans, France
[8]Instituto Nacional de Pesquisas Espaciais (INPE), São José dos Campos - SP, 12227-010, Brazil
[9]Instituto de Astronomia, Geofísica e Ciências Atmosféricas, São Paulo - SP, 05508-090, Brazil
[10]Physical Research Laboratory, Ahmedabad, Gujarat 380009, India
[11]India Meteorological Department, New Delhi, Delhi 110003, India
[12]NASA Langley Research Centre, Hampton, 23666, USA
[13]National Institute of Aerospace, Hampton, 23666, USA
[14]Indian Institute of Tropical Meteorology, Pune, Maharashtra 411008, India
[15]Institut für Atmosphäre und Klima, ETH Zürich, 8006, Switzerland
[16]Department of Earth and Environmental Sciences, Munich University, München, 80333 Germany
[17]Department of Chemistry, University of Waterloo, ON N2L 3G1, Canada

*Correspondence to*: Hazel Vernier (hazel.vernier@gmail.com)

**Abstract.** Volcanic eruptions are crucial in the Earth's climate system, driving natural variability. Typically, sulfate aerosols generated by major volcanic events have persisted for years, cooling the Earth's surface while warming the stratosphere. The unprecedented submarine eruption of Hunga Tonga Hunga Ha'apai (HTHH) on January 15th, 2022, challenged this "paradigm" and offered a new perspective by injecting material up to the mesosphere. It led to a 10% increase in the global stratospheric water vapor burden due to seawater injection, warming the Earth's surface and competing with the sulfate-induced cooling. The resulting Stratospheric Aerosol Optical Depth appears to be much lower than that derived from satellite observations based on previous eruptions. This study shows a unique combination of balloon-borne and satellite-based measurements of the HTHH water-rich aerosol plume. We used an innovative balloon-based sampling technique and ion chromatographic analysis of the collected aerosol samples to suggest the presence of $Na^+$, $K^+$, $NH_4^+$, $Ca^{2+}$, $Cl^-$, and traces of $SO_4^{2-}$, 8 months after the eruption, indicating its greater complexity than previously assumed.



Based upon the chemical and optical balloon-borne and satellite observations, we suggest that marine aerosols played a role in accounting for the larger-than-expected aerosol burden compared to the modest 0.42 Tg $SO_2$ injected. These findings encourage the inclusion of sea salt in addition to sulfate in climate models to correctly simulate the climatic impact of the Hunga eruption.

## 1 Introduction

The eruption of Hunga Tonga Hunga Ha'apai (HTHH), a submarine volcano, stands out for its direct injection of a massive plume of volcanic, and submarine material up to mesospheric levels, with a record-breaking altitude of ~57 km for the uppermost part of the plume (Proud et al., 2022; Khaykin et al., 2022). Unlike major eruptions with Volcano Explosivity Index (VEI) greater or equal to 6 (Table.1) are known for producing long-lasting sulfate aerosols from $SO_2$ emissions, which can cool surface temperatures for years (e.g., Mt. Pinatubo, Parker et al., 1996) a significantly lower amount of $SO_2$ (~0.42 Tg) was detected in the atmosphere. In contrast, a surprisingly large quantity of water vapor was measured (146 ± 5 Tg, ~10% of the total stratospheric burden) (Millán et al., 2022; Sellitto et al., 2022).

| Abbreviation | Full name | Erupted on | Lat, Lon | Injection altitude | VEI | Amount of $SO_2$ injected (Tg) |
|---|---|---|---|---|---|---|
| El | El Chinchon | 03/28/1982 | 17.3º N, 93.2° W | ~31 km | 5 | 7.5 |
| Pi | Pinatubo | 06/15/1991 | 15.14º N, 20.3°E | ~18-25 km | 6 | 20 |
| Ka | Kasatochi | 08/07/2008 | 52.17º N, 175.5º W | 14.5-16.5km | 3- 4 | 0.9 |
| Sa | Sarychev | 06/11/2009 | 48º N, 153.2º E | 15 Km | 4 | 0.9 |
| Na | Nabro | 06/12/2011 | 13.3º N, 41.7º E | 21 Km | 4 | 1.65 |
| Ca | Calbuco | 04/22/2015 | 41.3º S, 72.6º W | ~15-22 km | 4 | 0.35 |
| Ra | Raikoke | 06/22/2019 | 48.2º N, 153.2ºE | ~13-17 km | 4 | 1.5 |
| HTHH | HTHH | 01/15/2022 | 20.5º S, 175.3º W | 57 km | 5-6 | 0.45 |

**Table.1 Summary of high-magnitude volcanic eruptions corresponding to their $SO_2$ injections in the stratosphere.**

Satellite observations using the Aura Microwave Limb Sounder (MLS) revealed this unprecedented water vapor injection (Millán et al., 2022). The abundant water vapor injected into the atmosphere rapidly transformed $SO_2$ into sulfate aerosols, causing the plume to descend to 24-26 km within three weeks due to radiative cooling (Sellitto et al., 2022; Legras et al., 2022). Studies suggest that the injected $H_2O$ vapor may have halved the lifetime of $SO_2$ by accelerating its conversion to $SO_4$ aerosols (Zhu et al., 2022; Asher et al., 2023). Furthermore, satellite, ground-based lidar, and balloon-borne observations from Reunion Island documented unprecedented aerosol characteristics, including height, backscatter, and extinction coefficients, surpassing those observed since Mt. Pinatubo's eruption in 1991 (Asher et al., 2023; Taha et al., 2022; Baron et al., 2023). The abundant water vapor from HTHH also triggered chlorine activation, leading to rapid ozone depletion (Evan et al., 2023; Zhu et al.,



2023). Months following the eruption, satellite observations revealed a mid-stratospheric volcanic layer and a distinct $H_2O$ vapor layer positioned slightly above the aerosol layer at ~26 km (Schoeberl et al., 2022). Notably, the $H_2O$ vapor layer displayed a slow ascent (~0.044 km/day) driven by the Brewer-Dobson circulation, generating tropical gas tape recorders while the aerosol layer descended due to the gravitational settling of particles with a diameter estimated at ~1.2 µm (Schoeberl et al., 2022; Legras et al., 2022). The unprecedented water vapor injection following the HTHH eruption presents a new paradigm

for volcanic aerosol research. Its unique nature underscores the potential for long-term climate impacts. The significant increase in water vapor perturbation caused by the HTHH eruption may have contributed to surpassing the Earth's average temperature increase of 1.5°C above pre-industrial levels, a critical threshold outlined in the Paris Agreement climate negotiations (Jenkins et al., 2023). The significant warming effect is primarily attributed to the influence of water vapor, as the relatively small amount of $SO_2$ injected would have had a limited impact on its own (Zhu et al., 2022).

Nevertheless, recent radiative forcing estimates, incorporating observations of both stratospheric water vapor and aerosols, suggest a net cooling effect (Schoeberl et al., 2023). Despite the water vapor's increased downward infrared flux, the cooling effect from aerosols is expected to dominate, leading to a tropospheric cooling effect. This translates to an estimated 0.021°C cooler all-sky surface temperature in the southern hemisphere, consistent with previous predictions (Zhu et al., 2022; Sellitto et al., 2022). However, the HTHH eruption presents unique characteristics beyond its water vapor impact. The event resulted

in a significant influx of seawater into the stratosphere, along with unexpectedly high aerosol extinction coefficients (Khaykin et al., 2022). This study delves deeper into these unique characteristics by analyzing Stratospheric Aerosol Optical Depth (SAOD) and unique balloon-borne aerosol characterization within the plume.

## 2. Satellite and balloon measurements

The following section describes the satellite and balloon measurements used in this study.

### 2.1. Satellite Observations

### 2.1.1. CALIOP/CALIPSO

The Cloud-Aerosol Lidar with Orthogonal Polarization (CALIOP), onboard the Cloud-Aerosol Lidar and Infrared Pathfinder Satellite Observations (CALIPSO) platform, has observed aerosol and cloud layers of the Earth's atmosphere on a global scale between June 2006 and August 2023 (Winker et al., 2010). We used the CALIOP level 1 version 4.21 products and applied an

algorithm developed by Vernier et al. (2009) to calculate the Scattering Ratio (SR) at 532 nm in addition to a cloud filter for data with depolarization ratio lower than 5%. The data have been accumulated to create monthly mean SR latitudinal cross-sections.

### 2.1.2. SAGE III/ISS and GloSSAC

The Stratospheric Aerosol and Gas Experiment III (SAGE III), installed on the International Space Station (ISS), is used to study the stratosphere through solar occultation measurements. SAGE III/ISS is composed of an advanced spectrometer



designed to measure the transmission of sunlight through the Earth's atmosphere in ultraviolet and visible wavelengths, allowing for the determination of aerosol extinction, ozone, water vapor, and other trace gas profiles (Chu et al. 1999). SAGE III/ISS operates by scanning the limb of the Earth's atmosphere during solar and lunar occultations, offering a unique vantage point from the ISS's orbit. SAGE family of instruments, which operated since the late 70's, constitutes the main part of the Global Satellite-based Stratospheric Aerosol Climatology (GloSSAC) aiming to provide time series of stratospheric aerosol properties for chemistry and climate models (Thomason et al., 2018). Figure 1 shows the times of Stratospheric Aerosol Optical Depth (SAOD) from GloSSAC between 60° S-60° N with Raikoke and the Hunga eruption being the largest increase of SAOD since Mt. Pinatubo eruption. We note that the Stratospheric Aerosol Optical Depth (SAOD) was calculated from the tropopause to 40 km, not from the 380° K isentropic level (Khaykin et al., 2022). Given that the Hunga plume was injected directly into the stratosphere, while a significant portion of the Raikoke plume remained in the lowermost stratosphere (tropopause to 380° K), the overall SAOD values for both eruptions are similar. Therefore, the Hunga eruption did not produce a significantly larger SAOD than Raikoke, contrary to previous suggestions (Khaykin et al., 2022).

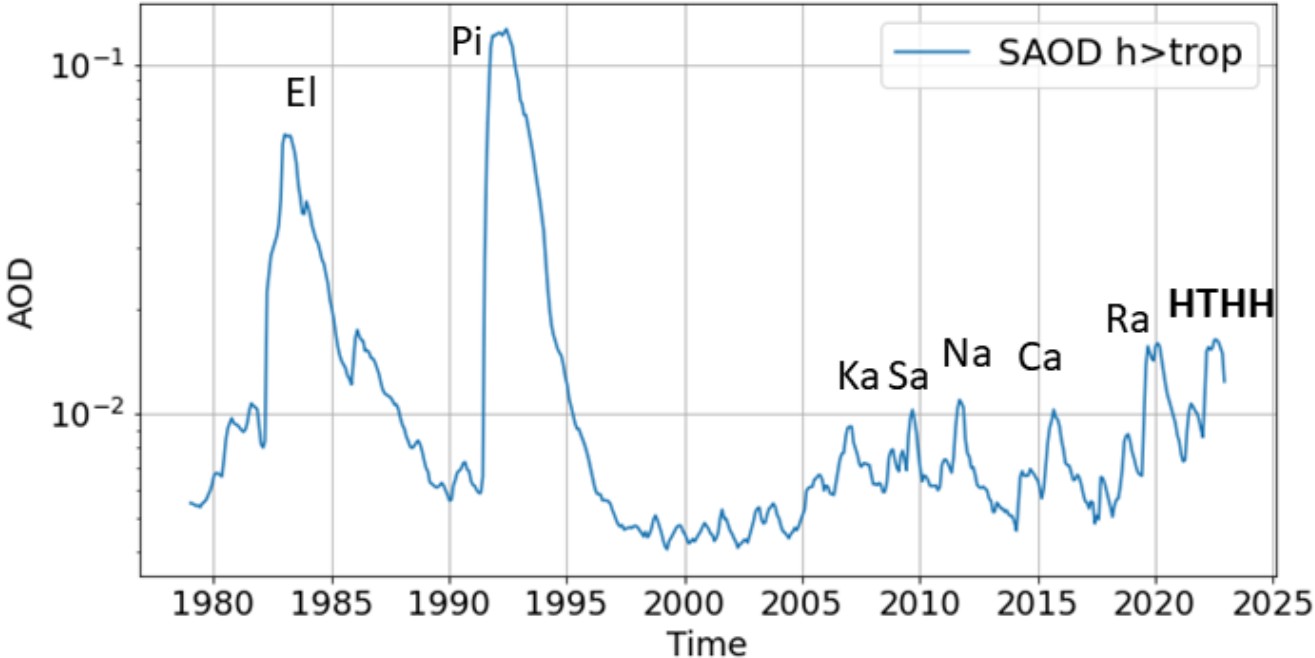

**Figure 1: Time series of global-mean Stratospheric Aerosol Optical Depth at 525 nm obtained from GLoSSAC database. El (El Chichon), Pi (Pinatubo), Ka (Kasatochi), Sa (Sarychev), Na (Nabro), Ca (Calbuco), Ra (Raikoke), HTHH (Hunga Tonga Hunga Ha'apai)**



## 3. BraVo campaign

The BraVo campaign involved deploying a suite of balloon-borne sensors in Bauru, São Paulo state (located at -22.36° S, -49.03° W), Brazil, to investigate the physical, chemical, optical, and microphysical properties of the HTHH aerosol layer. Ground-based lidar measurements conducted in both Sao Paulo and Bauru (approximately 400 km west) consistently detected

115    the aerosol layer in the mid-stratosphere, ranging from 17-25 km in altitude during the course of the campaign in August 2022 (Fig. 2).

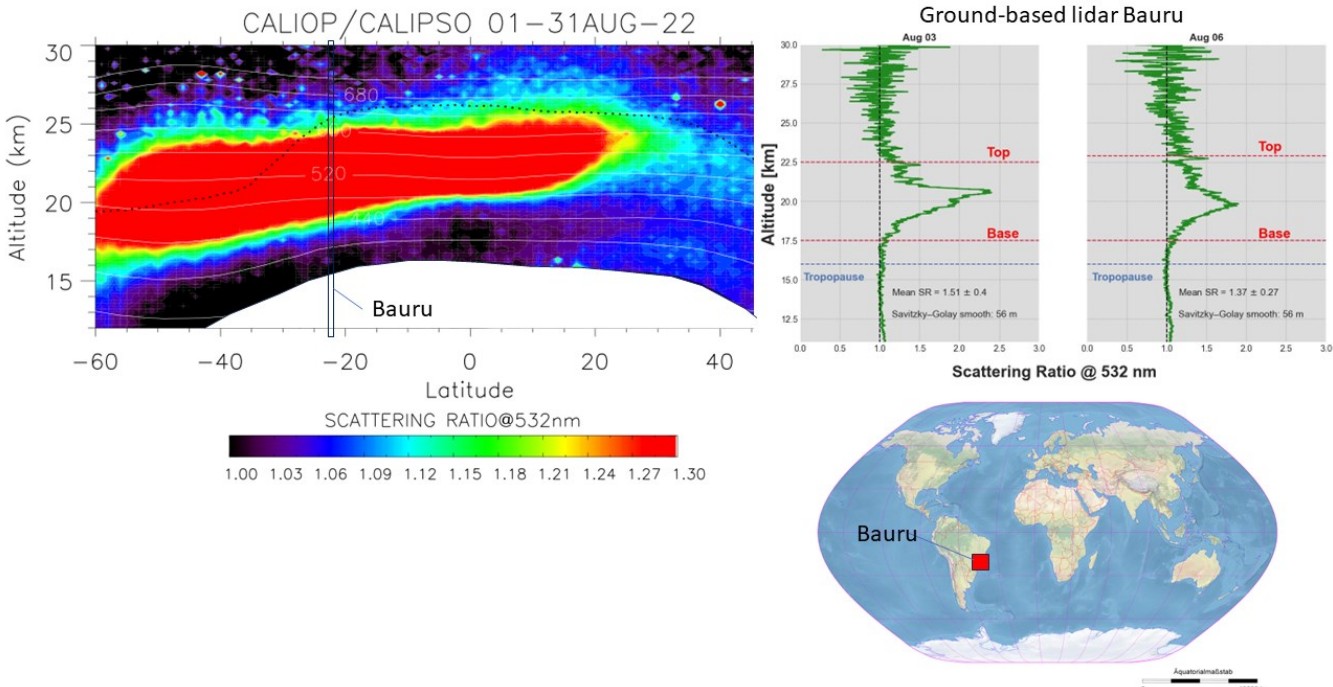

120    **Figure 2: (Top left) Latitudinal cross-section of mean scattering ratio (August 2022) showing the HTHH volcanic plume (red). (Top right) Ground-based lidar scattering ratio (532 nm) from Bauru (August 2022). (Bottom right) Map showing the BraVo campaign location in Brazil.**

These findings aligned with observations made by the Cloud-Aerosol Lidar and Infrared Pathfinder Satellite Observations (CALIPSO) satellite between 01-31 August 2022 (Fig. 2) even if some differences are observed especially at the top of the

125    plume which might be caused by the local versus zonally averaged satellite profiles. It is important to note that CALIPSO data were unavailable over Brazil due to interference from the South Atlantic Anomaly (SAA). The BraVo campaign utilized relatively low-cost balloon flights to study the Hunga plume directly within the mid-stratosphere, surpassing the limitations of conventional aircraft measurements. Notably, while previous balloon campaigns conducted in Reunion Island-France (Asher et al., 2023; Baron et al., 2023) and ground-based lidar measurements from Lauder (Khaykin, S., et al. 2022), New Zealand



investigated the HTHH plume, none focused on the chemical composition of the particles within it. Among other payloads, a crucial instrument deployed during the BraVo campaign for this study was the portable and lightweight CHEM filter sampler (Model-9401-FILT) (Dumelie et al., 2022). This self-contained unit is equipped with eight filter channels for efficient aerosol sampling and features onboard software, data storage, and remote-control capabilities for the sampling process. To extend the sampling duration within the HTHH plume, the sampler was coupled with a radio-controlled valve attached to the balloon's neck (Vernier et al., 2018). Additional information regarding the sampler and other payloads is provided below together with a list of all flights conducted in August 2022.

### 3.1. Instrument description (POPC, COBALD and Sampler)

### 3.1.1. POPC

A small balloon-borne Profiling Optical Particle Counter (OPC) (<2kg) was developed from the Particle Plus 8306 OPC (POPC) instrument for weather balloon applications. This POPC measures aerosol concentration profiles at eight radii between 0.15-5 μm from the ground to the stratosphere. The system employs a laser diode emitting at 785 nm. As particles enter the optical chamber, the scattered light at 90 degrees is amplified and converted into a voltage using a high-speed ADC (analog-to-digital converter), providing a digital measurement of the pulse height. A new version of the POPC, which utilizes a similar laser but with improved signal processing to derive size information for 30 channels between 0.15-5 μm was also used.

### 3.1.2. COBALD

COBALD is a lightweight (540 g) instrument that consists of two high-power light-emitting diodes (LEDs) emitting approximately 500 mW of optical power at wavelengths of 470 and 940 nm, respectively. The backscattered light from molecules, aerosols, or ice particles is recorded by a silicon photodiode using phase-sensitive detection. The precision of the backscatter ratio measurements is better than 1% in the UTLS region (Vernier et al. 2015).

Figure 3 presents total aerosol concentration profiles alongside the scattering ratio minus one for three flights conducted on August 7th, 12th, and 20th. Aerosol concentration peaks within the Hunga plume range from 8 to 9 #/cm³ at altitudes between 20.5 and 23 km. The corresponding variations in scattering ratio within the stratosphere show good qualitative agreement with the aerosol concentration profiles, demonstrating consistency between these independent measurements.





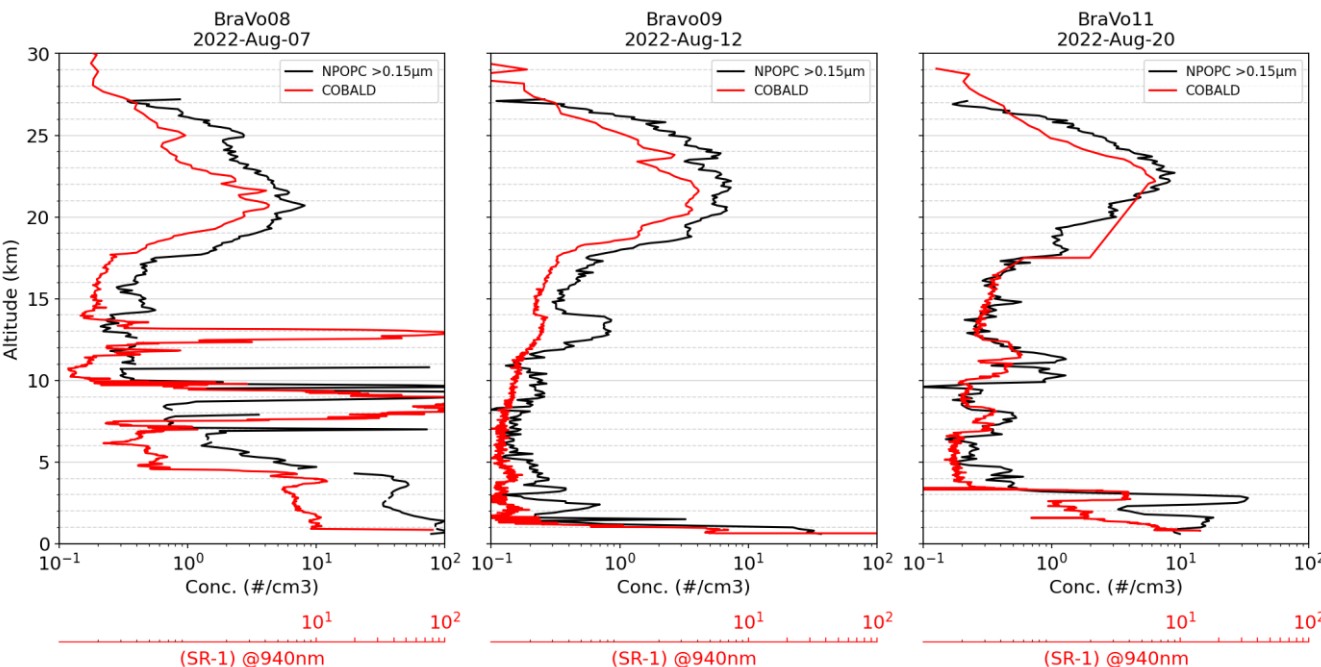

**Figure 3: (Above) POPC aerosol profiles (r > 0.15 µm) for August 7th, 12th, and 20th, along with COBALD aerosol backscatter ratio (backscatter-1) profiles at 940 nm for August 7th and 12th.**

**Sampler**

The CHEM filter Sampler (https://www.brechtel.com/product/filter-sampler/) is a simple device that captures aerosol particles on filters for offline chemical analysis. The sampler has eight filter cartridges that hold the individual filters. Using eight filters makes it possible to perform time-resolved sampling under user control. The CHEM filter sampler was adapted for balloon flight applications using an Imet radiosonde and a Raspberry PI system to control the sampling altitudes along the flight altitudes. The BraVo campaign employed a multi-pronged approach. Three sets of samples were collected on August 16th, targeting the boundary layer (Pos#2), the free troposphere (Pos#3), and the lower stratosphere (Pos#4) within the Hunga plume (Fig. 7). These samples were then compared with size-segregated aerosol concentration measurements obtained on August 7th,12th, and 20th using the Particle Plus Optical Particle Counter (POPC) instrument. This combined analysis provided valuable insights into the chemical composition and size distribution of the Hunga plume aerosols even if the measurements were not made on the same day. The valve slowed the balloon's rise by strategically releasing gas during ascent. This innovative approach allowed the sampler to capture the plume for over 40 minutes between 19 and 22 km, a significant improvement compared to the ~10-minute duration of traditional balloon flights. Internal parameters such as temperature, pressure, and pump flow rate were continuously recorded throughout the flight (Fig. 4). The sampler encountered temperatures ranging



between 17-36 °C during the flight. To ensure sample integrity, the samples were immediately stored in dry ice upon recovery. In a clean and controlled environment, the filters were unloaded in sterile petri dishes, sealed, individually labeled, and transported to France for IC analysis.

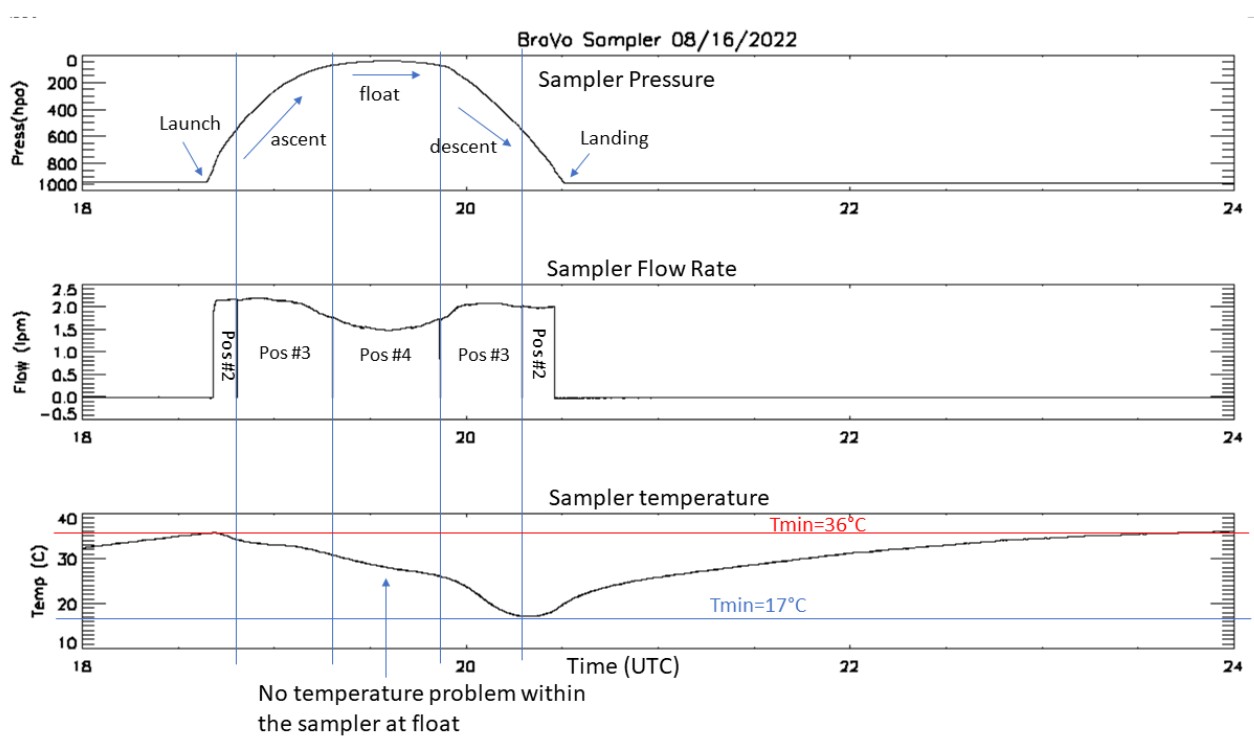

Figure 4: Sampler pressure, flow rate, and temperature during the August 16th flight. Vertical blue lines indicate sample positions (#2, #3, #4). Pressure data delineate flight phases (ascent, float, descent). Sampler internal temperature fluctuated between 17 °C and 36 °C.

**ECC/CFH**

The ECC measurement is based on the electrochemical oxidation of potassium iodide by ozone in an aqueous solution. Ozone oxidizes iodide ions, producing a measurable electrical current proportional to the ozone concentration. The accuracy of the ECC ozonesonde is typically ±5–10%.

The CFH measures water vapor concentration through the chilled-mirror principle using a cryogenic liquid as a coolant. A small mirror attached to the cold finger piece is alternately heated and cooled to maintain a thin layer of frost, which is optically detected. The mirror temperature, which is equal to the ambient dewpoint or frost-point temperature, is measured by a small





thermistor embedded in the mirror's surface. The frost-point temperature is then used to calculate the partial pressure of water vapor in the air and determine the water vapor mixing ratio (WVMR) with an uncertainty of ±4% in the lower tropical troposphere to ±10% in the middle stratosphere and tropical tropopause.

| Flight_ID | Launch Time UTC | CFH | ECC | POPC | COBALD | Sampler | SAGE III/ISS |
|---|---|---|---|---|---|---|---|
| 20220807_BRU_BraVo_08 | 6h30 | x | x | x | x | | |
| 20220812_BRU_BraVo_09 | 6h30 | x | x | x | x | | x |
| 20220816_BRU_BraVo_10 | 18h30 | | | | | x | |
| 20220820_BRU_BraVo_11 | 6h30 | x | x | x | x | | |


**Table.2 List of flights conducted during the BraVo campaign in August 2022**

### 3.2 Radio-controlled Balloon System

Previous balloon-borne aerosol collection efforts were hindered by insufficient mass acquisition. To address this, we successfully implemented a radio-controlled balloon system (Vernier et al., 2018) during this campaign. This system, which utilizes a radio-controlled valve to strategically vent gas, significantly extended the flight duration within the plume, enabling the filtration of a larger aerosol mass. The system underwent prior testing during field campaigns conducted in India.

Due to unfavorable weather and logistical constraints, we were limited to a single flight with this enhanced system. Weight restrictions prevented the co-deployment of the POPC on this specific flight. However, POPC measurements were successfully obtained four days before the controlled ascent, as well as on August 7th and August 20th (Fig. 3).

Figure 5 (left panel) illustrates the GPS altitude, sampling time, and temperature profile of the controlled balloon flight. The right panel presents the total aerosol concentration for particles exceeding a specific diameter, as measured by the POPC during separate flights. Lidar and POPC observations consistently indicated a heterogeneous plume distribution throughout the campaign. The HTHH plume was situated between 18 and 26 km, with larger aerosols (diameter > 0.8 µm) concentrated near the base (18-20 km) and smaller particles dispersed more widely up to 25 km. The controlled flight trajectory included a deliberate reduction in ascent rate at 19.5 km, followed by a slow climb to a peak altitude of 21.6 km before descent. This maneuver, combined with the extended float period, allowed the aerosol sampler to remain within the plume layer for ~40 minutes significantly longer than a conventional balloon flight, maximizing aerosol collection.



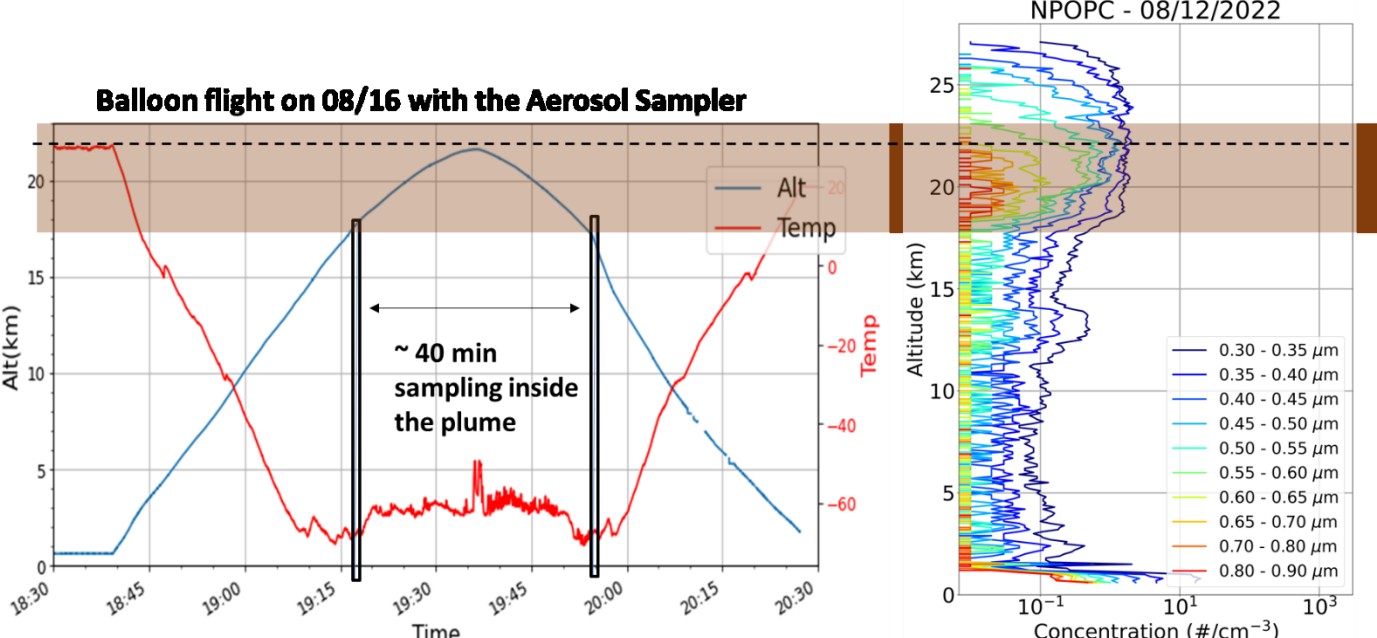


**Figure 5: (Left) Trajectory of the radio-controlled balloon flight showing GPS altitude and temperature (Right) Aerosol concentration profiles from POPC show that the sampling was done within the HTHH plume.**

## 3.3 Offline analysis and Ion Chromatography


Two Dionex Ion Chromatographs (Integrion HPIC) were used to analyze water-soluble inorganic species (WSIS) such as $K^+$, $Na^+$, $Mg^{2+}$, $Ca^{2+}$, $NH_4^+$, $Cl^-$, $F^-$, $Br^-$, $NO_2^-$, $NO_3^-$, $SO_4^{2-}$, and $PO_4^{3-}$ using conductivity detection. AS-18-FAST-4 µm, 2x150 mm was used as the anion column with a KOH gradient as eluent for 25 minutes: 10mM from 0 to 6 min, 10 mM to 60 mM from 6min to 15min, and 60 mM from 15min to 25min. CS-12A, 4x2500 mm was used as the cation column with Methyl Sulfonic

Acid gradient as eluent for 25 minutes: 10 mM from 0 to 6min, 10 mM to 35 mM from 6min to 15min, and 35 mM from 15min to 25min. An EQ7000 Millipore ultrapure water purification system (Resistivity ≥18.2 MΩ.cm) was used to generate ultrapure water used for dilutions, extractions, and preparation of the mobile phase columns. A 5 mg/L mix was prepared from a 1000 mg/L stock solution of each cation using a highly pure analytical grade standard solution from Merck ($NH_4Cl$, $CaCO_3$, $KNO_3$, $Mg (NO)_2$, and $NaNO_3$) for calibrating each cation. Likewise, high purity grades of anion standards from Merck (NaCl, $NaNO_3$,

$NaBr$, $NaNO_2$, and $NaSO_4$) were prepared for calibrating each anion. The cation, and anion mix were diluted, to create a series of working standards with concentrations of 1, 5, 10, 50, 100, and 500 µg/L. These were injected into the instrument (sampling loop of 250 µL) to create a calibration range for the samples. The ultrapure water was pre-checked for probable cation, & anion contamination before sample analysis. The detection limit of the IC used was estimated to be 12.8 ng/6ml, indicating that we needed to collect 12.8 ng of each ion on those filters. Filters on positions #5 to #8 of the sampler were used as "true



blanks". We followed the extraction procedure that was previously described (Vernier, J-P. et al., 2018; Vernier, H. et al., 2022). Each sample was injected thrice. The average values of each ionic species were calculated and the data was analyzed using the same method described earlier ((Vernier, J-P. et al., 2018; Vernier, H. et al., 2022). The analysis involved three rounds of sample measurement. We calculated the mean value and standard deviation. We then used this standard deviation to account for the error when averaging the samples collected from positions #2, #3, and #4 (excluding the "blank" values).

This approach essentially subtracted the blank value from these measurements. The error bar in Fig.7 reflects this error estimate, incorporating the standard deviation after blank subtraction.

## 4. High Aerosol Optical Depth Despite Low SO₂ Injection: A Unique Case of HTHH

Explosive volcanic eruptions inject $SO_2$ into the stratosphere, where it undergoes oxidation and nucleation, transforming into

sulfate aerosols (Robock, A., 2000). These aerosols play a crucial role in Earth's radiative budget by reducing incoming shortwave radiation reaching the surface while absorbing near-infrared radiation and trapping outgoing longwave radiation. This combined effect leads to cooler surface temperatures and a warmer stratosphere (Robock, A., 2000). Furthermore, the altered radiative budget can trigger a cascade of global climate effects, including a suppressed global water cycle ((Robock, A., 2000); Zuo et al., 2022), anomalous winter warming over Eurasia ((Robock, A., 2000); Stenchikov et al., 2002), El Niño-

like sea surface temperature responses (Zuo et al., 2018; Sun et al., 2019), and a weakening of monsoon circulation. These findings highlight the complex interplay between volcanic eruptions, aerosol formation, and global climate dynamics.





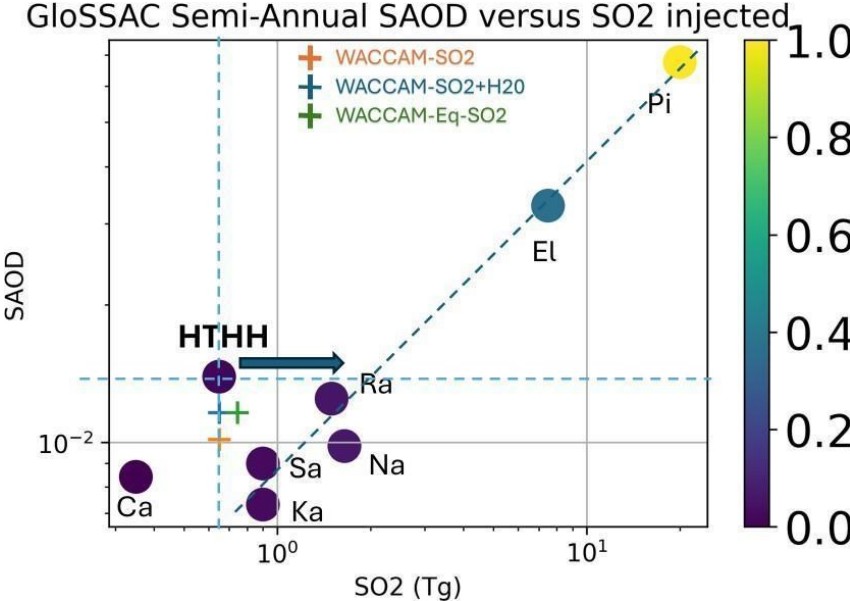

**Figure 6: Global Stratospheric Aerosol Optical Depth (SAOD) and Volcanic Influences Scatter plot of semi-annual SAOD retrieved**
**from the Global Space-based Observing System of Aerosols and Clouds (GloSSAC) between 1979 and 2023 as a function of estimated**
**SO2 injected. SAOD values are derived from a combination of satellite observations above the tropopause. Colored circles depicting**
**peak volcanic SO2 emissions from major eruptions (El Chichon - El 03/1982, Mount Pinatubo - Pi 06/1991, Kasatochi - Ka 08/2008,**
**Sarychev - Sa 06/2009, Nabro - Na 06/2011, Calbuco - Ca 04/2015, Raikoke - Ra 06/2019, and Hunga Tonga-Hunga Ha'apai - HTHH**
**01/2022) versus the corresponding semi-annual mean (sSAOD) (averaged over six months following the eruption). Model simulations**
**of the HTHH eruption from WACCM with and without water injected are shown with the colored crosses.**

To investigate the link between volcanic sulfur dioxide (SO₂) injections and their impact on stratospheric aerosol burden over
the past four decades, we used data from the Global Space-based Stratospheric Aerosol Climatology (GloSSAC) database
(Kovilakam et al., 2022) (Figure 6). GloSSAC was specifically designed to provide consistent, long-term datasets, enabling
the creation of reliable observational records for stratospheric aerosols. It's important to note that selecting a consistent latitude
range for both eruptions (0-80° N for Raikoke and 60° S-0° for HTHH) resulted in comparable Stratospheric Aerosol Optical
Depth (SAOD) values. Since sulfate aerosols formed from volcanic SO₂ are considered the main driver of SAOD and
subsequent radiative and climate impacts (Carn et al., 2022). Figure 6 presents a scatter plot illustrating the relationship
between the sulfate SAOD (sSAOD) following major eruptions and the corresponding volcanic SO₂ injection. The HTHH
eruption stands out. Despite injecting a relatively modest amount of SO₂ (0.42-0.5 Tg) (Carn et al., 2022), it produced a
significantly higher sSAOD than moderate eruptions observed since the late 2000s (Fig. 1). For example, the Raikoke eruption
in 2019 injected about 1.5 Tg of SO₂ but resulted in an sSAOD of 0.012, while HTHH with its lower SO₂ injection produced
an sSAOD 16% higher. Compared to the two largest eruptions (8 Tg for El Chichón and 20 Tg for Mt. Pinatubo) with direct
stratospheric injection, the HTHH eruption exhibited a remarkably lower SO2/sSAOD ratio. This anomaly suggests that the



HTHH plume may have contained non-sulfate aerosols. The HTHH eruption resulted in distinct aerosol particle properties in two layers: near the tropopause and in the mid-stratosphere (Kahn et al., 2024). The Multi-angle Imaging SpectroRadiometer (MISR) instrument on NASA's Terra satellite detected only spherical, non-light absorbing particles for a week. Near-tropopause particles had relatively constant sizes, while mid-stratosphere particles were smaller but grew downwind (Kahn et al., 2024).

Model simulations from the Whole Atmosphere Community Climate Model (WACCM) (Zhu et al., 2022) were also investigated to simulate sSAOD through 3 different simulations: 1) Without volcanic emissions, 2) with only $SO_2$, and 3) with $SO_2$ and $H_2O$ (Fig.1). With $SO_2$ only, sSAOD derived from the model is lower by 27% than GloSSAC and the simulation with $H_2O$ increases the sSAOD by 10%. Indeed, the effect of $H_2O$ on the plume evolution is to halve the lifetime of $SO_2$ (Zhu et al., 2022) by producing additional OH radicals for the oxidation reaction but with relatively limited impact on the resulting

maximum of SAOD. The high SAOD could be due to enhanced extinction caused by water vapor-induced swelling of existing aerosols (Asher et al., 2023) which nevertheless does not appear to be a major factor in WACCM. While the initial plume analysis revealed a 30% increase in particle size likely caused by the high water vapor content (Asher et al., 2023), this effect likely diminished over time as the water vapor dispersed. New aerosol/$SO_2$ retrievals using the TROPOMI instrument on Sentinel 5P (Krotkov et al., 2024) indicate that a very significant fraction of sulfate was already present in the plume (0.6 Tg)

2 days after the eruption on January 17th coexisting with ~0.5 Tg of $SO_2$, similar to previous estimates (Carn et al., 2022). Therefore, unaccounted sulfate production within the early eruptive column could contribute to the unusual sSAOD observed and indicate that the initial SO2 injection is largely underestimated.

   To study this unprecedented volcanic event, the Brazil volcano (BraVo) project was launched.








# 5. Results


## 5.1 Chemical analysis

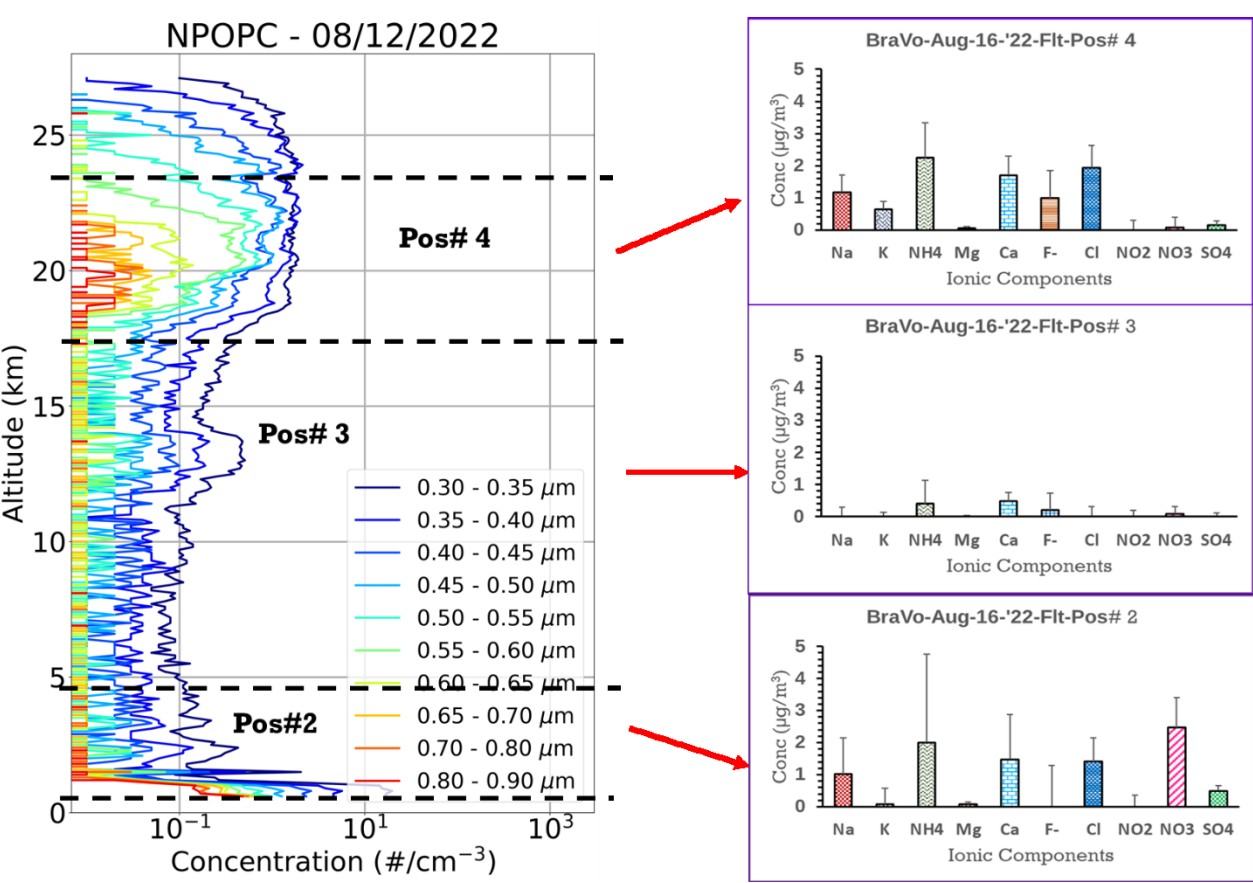


**Figure 7: (Left) POPC results of the aerosol concentration profile measured on Aug. 12th. (Right) IC results of the samples collected during the flight on 16th Aug. Flight samples were collected at three distinct layers: boundary layer (Pos #2), free troposphere (Pos #3), and lower stratosphere within the HTHH plume (Pos #4). Error bars represent the standard deviation calculated from three repeated IC analyses for each identified species.**


Aerosol mass measurements at Pos #4, within the HTHH plume, were significantly higher ($9 \pm 1.8$ µgm⁻³ STP) than those in the free troposphere (Pos #3; $1.2 \pm 0.6$ µgm⁻³). The plume sample at Pos #4 exhibited a diverse ionic composition, including $Na^+$, $K^+$, $NH_4^+$, $Ca^{2+}$, $Cl^-$, and traces of $SO_4^{2-}$. This suggests a potentially complex mixture within the plume, possibly influenced by various sources (e.g., volcanic emissions, and seawater entrained). However, the presence of F⁻and $NO_3^-$ requires further



investigation due to the high associated error bars. In contrast, the free troposphere (5-17 km, Pos#3) displayed a considerably
lower total ionic concentration ($1.2 \pm 0.6$ µgm⁻³), with $Ca^{2+}$ as the only consistently detectable ion. The boundary layer sample
(1.6-5 km, Pos #2) contained a variety of ions, including $Na^+$, $K^+$, $NH_4^+$, $Mg^{2+}$, $Ca^{2+}$, $Cl^-$, $NO_3^-$, and $SO_4^{2-}$. The total ionic mass
in this layer was measured to be $9.4 \pm 4.3$ µgm⁻³. However, due to higher uncertainties associated with some ions, only $Ca^{2+}$,
$Mg^{2+}$, $NO_3^-$, and $SO_4^{2-}$ are considered reliable for further analysis. Several studies have reported a positive correlation between

$NO^-$ and mineral dust, suggesting the presence of aerosol $NO^-$ incorporated within dust particles (Fairlie et al., 2010). Notably,
only $NO_2^-$ was present in reportable quantities at Pos #4. Due to associated error bars, other ions measured at this position
were not statistically significant and are not reported here. The observed Cl:Na ratio of 1.5 at Pos #2 aligns closely with that
of standard seawater (1.8), further supporting the potential presence of sea salt (NaCl) at this location. Back trajectory analysis
indicates that the air masses sampled at Pos #2 originated from the Eastern Pacific but were influenced by South American

landmasses, particularly the southern Amazon region, during the preceding 4-5 days. This suggests potential exposure to
wildfire smoke from these regions (Fig. 8). To visualize the plume localization, we overlaid a map of DeepBlue AOD (from
MODIS) for the period between August 11th and August 16th with the 3000m back trajectories. While the majority of the air
masses originated from South America, it's also possible that some air parcels could have originated from the Eastern Pacific,
potentially carrying traces of sea salt 5 days prior to our measurements.



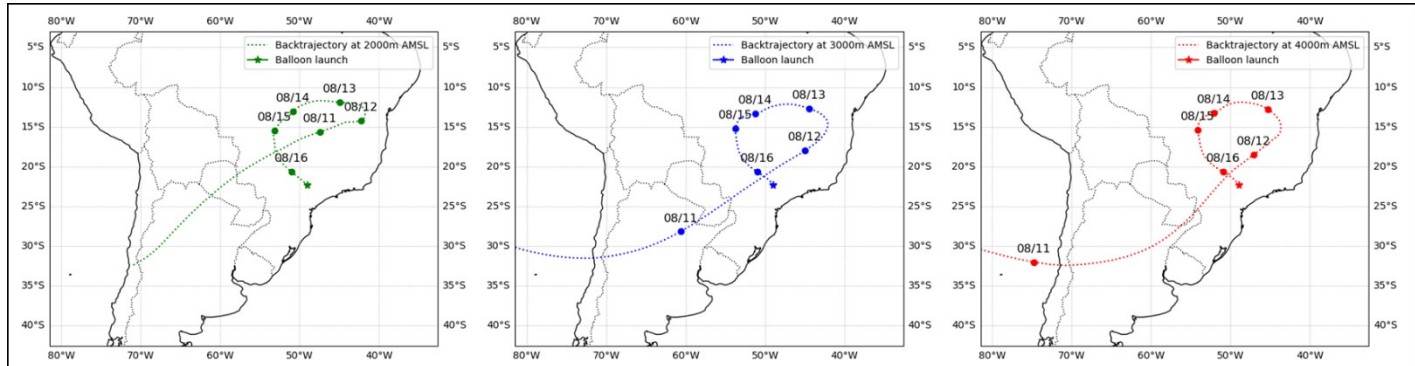

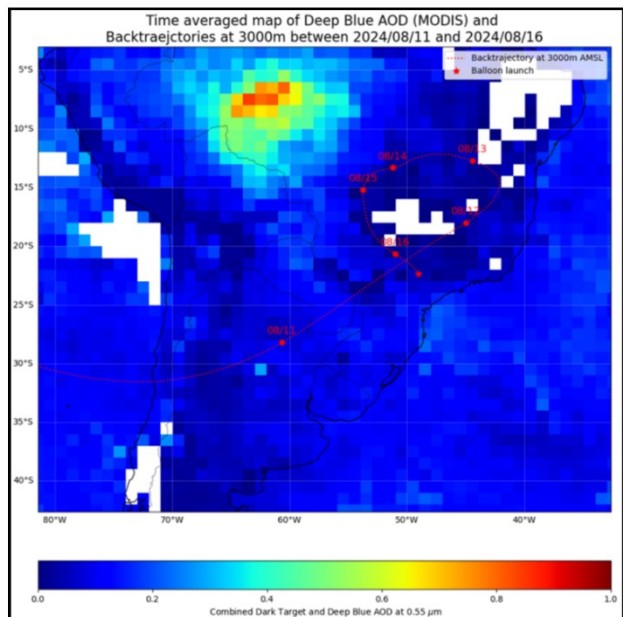


**Figure 8: (Top Row) Back trajectory analysis at 2000m (left), 3000m (center), and 4000m (right) AMSL for air masses sampled during the BraVo campaign. Markers indicate air mass location dates, tracing back from the balloon launch site. These trajectories (1.6-5 km) suggest air masses originated south of high AOD regions, indicating limited wildfire smoke exposure during sampling. (Bottom) Time-averaged MODIS Deep Blue AOD map (August 11-16, 2024) with overlaid 3000m back trajectories. High AOD areas**

**may indicate wildfire smoke. Air masses primarily originated from South America, with some from the Eastern Pacific, potentially carrying sea salt from 5 days prior. The color bar indicates AOD values.**

## 5.2 POPC measurements reveal aerosol size distribution

POPC measurements conducted on August $7^{th}$, $12^{th}$, and $20^{th}$ provide valuable insights into the size distribution of aerosols

within the HTHH plume (Fig. 9). The figure presents differential concentration profiles for ten size bins of particles ranging from 0.3 μm to 0.9 μm in diameter, based on data from three separate flights. Size distribution measurements were conducted for two altitude ranges: 19-21 km and 21-24 km. All flights consistently detected a layer of larger aerosols (diameter exceeding





0.8 µm) concentrated near the plume's base (19-21 km). However, the peak concentration for smaller particles (diameter exceeding 0.3 µm) varied, with peaks observed around 21 km on August 7th and 23 km on August 20th (Fig.9). POPC data suggest vertical variations in aerosol size distribution within the Hunga Tonga-Hunga Ha'apai (HTHH) plume, with larger aerosols concentrated at lower altitudes. Volumetric size distribution data from 19-21 km and 21-24 km consistently showed a peak near 0.4-0.6 µm. Further details on POPC measurements and comparative analyses with COBALD are presented in the next section. Figure 10 includes quantitative comparisons with collocated SAGE III/ISS extinction profiles. POPC extinction profiles were derived using composition-based correction factors, log-normal distribution fitting, and Mie theory. Assuming sulfate (RI=1.42) or sea salt (RI=1.50) aerosols improved agreement between SAGE III/ISS, COBALD, and POPC (455-1020 nm). However, due to similar refractive indices, the comparison does not rule out the presence of sea salt-like aerosols in the plume.

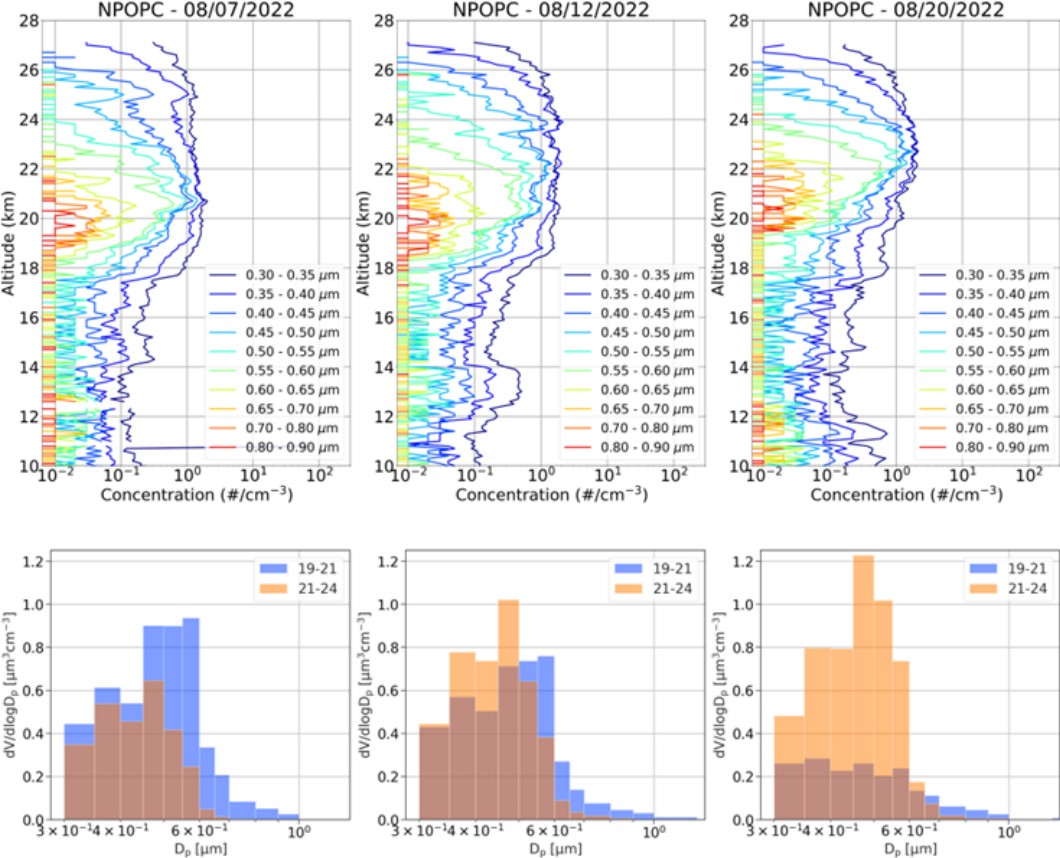

**Figure 9: (Top) Aerosol concentration profiles for 10 particle size bins (from 0.3 µm-0.8 µm) obtained from three POPC flights conducted on August 7th, 12th, and 20th. (Bottom) The corresponding volumetric size distribution is shown for two altitude ranges: 19-21 km and 21-24 km. The dark orange color represents the combined size distribution where two distributions overlap each other. The plume was not fully homogeneous during this period and displayed day-to-day variability.**



**6. Deriving aerosol extinction and quantifying aerosol mass with POPC; Comparison to Satellite Observations (SAGE III/ISS)**

Our study utilized a commercially available Optical Particle Counter (OPC) sensor modified for high-altitude balloon measurements. This modified Profiling Optical Particle Counter (POPC) incorporates a flow control system and telemetry interface via an IMet radiosonde and measures aerosol particle number concentrations across 30 size channels ranging from 0.3-10 μm. Since 2020, it has been successfully deployed in various campaigns across India, the US, and France (Vernier, J-P. et al., 2018; Dumelié, N., et al). The POPC size thresholds are determined using known PolyStyrene Latex (PSL) particles. Figure 10 shows the Scattering Efficiency for individual particles derived from Mie calculations (Hagan, D. H. and Kroll, J. H.(2022) for PSL, Sulfate ($70\%H_2SO_4/30\%H_2O$) (Knepp et al., 2024), and Sea Salt (Bi et al., 2018) by accounting for the physical parameters of the POPC (laser wavelength, detector position, size thresholds). The scattering efficiency for both sea salt and sulfate is systematically lower than PSL and therefore the size of the POPC within the HTHH plume is underestimated. A correction factor was applied to the size threshold of the POPC to account for the lower scattering efficiency of particles that may have been present in the HTHH plume compared to the PSL particles.

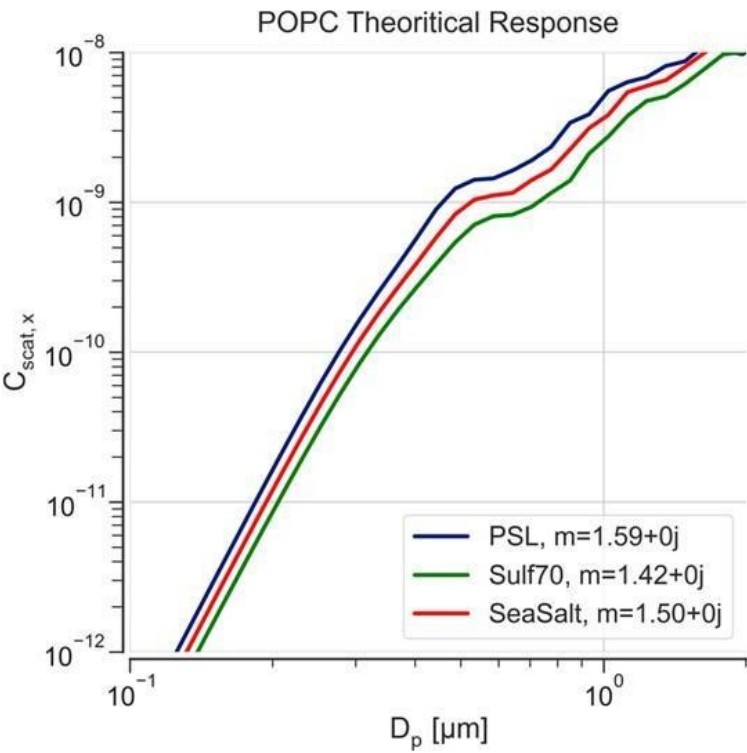

**Figure 10: Mean Scattering Efficiency for three different types of particles (PSL, Sulfate, Sea Salt) near 785 nm**



The systematic lower scattering efficiency of sulfate and sea salt led us to test how the assumption on composition could affect the retrieved extinction profiles and theoretical mass (Bi et al., 2018) compared to SAGE III/ISS, COBALD, and the sampler data on 08/16. The balloon flight with POPC and COBALD was launched to coincide with a SAGE III/ISS occultation measurement performed within a 300 km radius and a 2 hr window. Fig. 11 provides further details on POPC measurements and comparative analyses with the Compact Optical Backscatter Aerosol Detector (COBALD) for these flights. Quantitative comparisons in Fig. 11 also include collocated extinction profiles from the Stratospheric Aerosol and Gas Experiment (SAGE) III. POPC extinction profiles were calculated using composition-based correction factors, log-normal distribution fitting, and Mie theory at various wavelengths (see Methods). The assumption of sulfate (RI=1.42) or sea salt (RI=1.50) aerosol composition yielded better agreement between SAGE III/ISS, COBALD, and POPC data across the 455-1020 nm spectral range.

The corrections (Sul70, Sea Salt) improved the comparison between SAGE III/COBALD and POPC between 18 and 26 km within the Hunga plume. Assuming sea salt improves especially the extinction profile at 1022 nm, which becomes, for most of the profile, within +/- 50% from SAGE III. SAGE III and OPC measurements over Laramie are within +/-40% for most of the cases in background conditions (Deshler,T. et al., 2003). Obtaining such an agreement with the POPC within the HTHH plume indicates the robustness of the POPC measurements in volcanically perturbed conditions. Table 3 provides a summary of the overall comparison after calculating the SAOD from the tropopause to 40 km. Overall, correcting the POPC improves the comparison with SAGE III. While the sea salt assumption tends to improve the comparison in the near InfraRed, the sulfate assumption seems better in visible light. Given the similar optical properties of sea salt and sulfate, the comparison remains inconclusive regarding the composition of the plume we can derive from this comparison and therefore we cannot rule out the presence of sea salt-like aerosols in the HTHH plume.





**Figure 11: POPC-derived, SAGE III/ISS, and COBALD Extinction profiles on 08/12. a) &b) no correction, c) & d) size corrected assuming Sulfate Refractive Index=1.42, e) & f) assuming Sea Salt Refractive Index=1.50. The POPC data fitted with a lognormal distribution before deriving the extinction values**

The theoretical mass from the POPC was calculated by using the flow rate of the sampler and the ascent rate of the flight on 08/16 together with the aerosol mass concentration derived from POPC corrected size distribution after assuming an aerosol density of 1.6 g cm$^{-3}$. The POPC data were fitted using a lognormal distribution before calculating the mass concentration. The



theoretical mass calculated using POPC measurements on 08/12 is closer to the observed mass if we assume the presence of
sulfate (within 10%) but overestimated by 53% for sea salt.

| Variables | Sampler | SAGE III | COBALD | POPC | POPC_Sul70 | POPC_Seasalt |
|-----------|---------|----------|--------|------|------------|--------------|
| SAOD_521 | x | 0.022 | x | 0.011(-50%) | 0.023(4%) | 0.027(18%) |
| SAOD_1022 | x | 0.010 | x | 0.002(-80%) | 0.007(-0%) | 0.010(0%) |
| SAOD_455 | x | x | 0.025 | 0.011(-56%) | 0.023(-8%) | 0.021(-16%) |
| SAOD_940 | x | x | 0.008 | 0.002 | 0.007(-12%) | 0.008(0%) |
| Mass($\mu$/m3) | 9.8 | x | x | 3.3(-66%) | 10.7(10%) | 15.0(53%) |

**Table.3 Comparison between SAOD at different wavelengths (column 1; line 1-4) calculated from POPC without (column "POPC")**
**corrections assuming either sea salt (column "POPC_SeaSalt") or Sulfate (column "POPC_Sul70"). In brackets are the percentage**
**differences between SAGE III and COBALD SAOD. The last line compares the mass concentration from the samplers and those**
**derived from the POPC.**

**6. Discussion**

**6.1 Complex composition in the HTHH plume**

Our analysis of aerosol samples collected on August 16th in Bauru, Brazil, using a lightweight balloon-borne aerosol sampler,
reveals a diverse ionic composition (Fig. 8). Pos #4, located within the HTHH volcanic plume as determined by additional
aerosol balloon data, exhibited notable concentrations of $Na^+$, $K^+$, $NH_4^+$, $Ca^{2+}$, and $Cl^-$, with traces of $SO_4^{2-}$ indicating the
possible presence of marine and volcanic aerosols. The subsequent discussion will examine these findings in more detail.
Newly formed sea salt aerosols have similar compositions to seawater, mainly containing $Na^+$, $Cl^-$, $Mg^{2+}$, S, $Ca^{2+}$, $Br^-$, and
$K^+$(Adachi, K., & Buseck, P. R. (2014). Sea salt aerosols form as seawater evaporates, increasing the concentration of salts.
Different salts become insoluble at different levels of saltiness, causing them to precipitate and form crystals. The amounts of
$Na^+$, $K^+$, and $Ca^{2+}$ in aerosols are influenced by how easily these salts can move from seawater into the air. Cl/Na ratio found
on Pos #4 of 0.6 is similar (within 25%) to what is expected in sea salt (0.8). Energy Dispersive X-ray and IC analysis of ash
particles collected on the ground less than two weeks after the HTHH eruption confirmed the presence of NaCl (Colombier et
al., 2023). While significant amounts of salts were deposited on land, attached to volcanic ash, it's probable that a substantial
quantity of salt and aerosol particles, unattached to ash, persisted in the atmosphere post-eruption. Previous research raised
questions about the origin of a hazy layer observed at the top of the plume (Proud et al., 2022). Similar haze has been linked
to sea salt emissions from littoral eruptions at Kilauea volcano (Woodcock, A. H. & Spencer, A. T. (1961). Recent studies
suggest that water-soluble compounds, including iodine, bromine, and chlorine, may have been injected into the stratosphere




in substantial amounts by the HTHH eruption(Vömel et al., 2022), potentially impacting stratospheric ozone chemistry.
Volcanic eruptions release gases such as halogens and $SO_2$, which are removed from the atmosphere through processes like
adsorption onto ash particles and deposition. While ash particles can accelerate gas removal, the specific mechanisms remain
unclear (Marshall et al., 2022). Eight months post-eruption, our analysis of the Hunga Tonga-Hunga Ha'apai (HTHH) plume
suggests potential sea salt injection into the stratosphere. Using equations from Zhang et al. (2021), we calculated sea salt
concentrations (Table 4) based on measured $Na^+$ and $Cl^-$ levels, which are well-established seawater tracers.

| Sampler Position | $Cl^-$ (µg/m3) | $Ca^{2+}$ (µg/m3) | $K^+$ (µg/m3) | $Mg^{2+}$ (µg/m3) | $SO_4^{2-}$ (µg/m3) | $Na^+$ (µg/m3) |
|---|---|---|---|---|---|---|
| Pos# 2 | 1.4 | 1.48 | 0.07 | 0.07 | 0.49 | 1.01 |
| Pos# 3 | - | 0.49 | - | 0.01 | 0.01 | - |
| Pos3 4 | 1.9 | 1.71 | 0.23 | 0.4 | 0.15 | 1.18 |


| Sampler Position | ss-$Cl^-$ (µg m$^{-3}$) | ss-$Ca^{2+}$ (µg m$^{-3}$) | ss-$K^+$ (µg m$^{-3}$) | ss-$Mg^{2+}$ (µg m$^{-3}$) | ss-$SO_4^{2-}$ (µg m$^{-3}$) | Sea salts (µg m$^{-3}$) |
|---|---|---|---|---|---|---|
| Pos# 2 | 1.82 | 0.04 | 0.04 | 0.12 | 0.25 | 3.02 |
| Pos# 3 | - | - | - | - | | - |
| Pos# 4 | 2.12 | 0.04 | 0.04 | 0.14 | 0.30 | 3.83 |

**Table.4 (Above) The concentration of $Cl^-$, $Ca^{2+}$, $K^+$, $Mg^{2+,}$ and $SO_4^{2-}$ in the BraVo samples analyzed with IC. (Below) sea salt concentration based on the measured $Na^+$ and $Cl^-$ levels using equations by Zang et al., 2021.**

The results indicate that the concentrations of $Cl^-$ and $Mg^{2+}$ in Pos #2, as well as $Cl^-$ and $SO_4^{2-}$ in Pos #4, are higher than
expected based on seawater composition alone. While $Na^+$ is often used as a reference element for sea salt, it's possible that
additional sources contributed to the slightly higher concentrations of other ions (ss-X) in our samples where X denotes $Cl^-$,
$Ca^{2+}$, $K^+$, $Mg^{2+}$, and $SO_4^{2-}$. This could have led to an overestimation of the total sea salt concentration. We believe that the
combination of our findings, including the elevated $Na^+$ and $Cl^-$ levels strongly supports the presence of sea salt aerosols in
the HTHH plume.

The Ca/Cl ratio was measured to be 0.8, significantly higher than the seawater value of 0.02 (see Table 5). The high Ca/Cl
ratio found in our samples could be due to the additional source of $Ca^{2+}$ other than seawater alone. The excess $Ca^{2+}$ observed
in aerosol samples may originate from the weathering of volcanic material on the seafloor, hydrothermal alteration of oceanic
basalts by seawater, and significant riverine input (Humphris, 1972). Magma composition, dominated by Ca-rich minerals
like olivine and pyroxene in mafic and ultramafic rocks, can significantly impact seawater chemistry following eruptions.



Additionally, $Ca^{2+}$ could potentially be associated with gypsum ($CaSO_4 \cdot 2H_2O$) and/or bassanite($2CaSO_4 \cdot \frac{1}{2}H_2O$). Ground samples collected shortly after the Hunga Tonga eruption exhibited a near 1:1 molar ratio of Ca:SO4 and Na:Cl, suggesting the presence of both calcium sulfate and sodium chloride (Colombier et al., 2023). This ratio could arise from two primary
mechanisms: (1) the direct interaction of seawater with volcanic ash, leading to the deposition of these salts, or (2) the formation of CaSO4 and NaCl through reactions between volcanic ash and gases like SO2 and HCl. While both processes are plausible, the observed ratio alone cannot definitively distinguish between them. Although significant time elapsed between the ground sample collection and our stratospheric measurements, the similarity in the presence of these salts suggests potential connections in the initial formation processes.

However, $Mg^{2+}$ appears to be at a very low concentration on pos #4 while it's among the major inorganic components in seawater (Millero, F. J. et al., 2007). When lava comes in contact with seawater, the initial reaction that occurs is the removal of $Mg^{2+}$ to create Mg(OH) -silicate phases, with the release of $H^+$ to solution (Resing, J. A., and Sansone, F. J. (1999) where the net reaction is: $Mg^{2+} + 2H_2O \rightarrow Mg(OH)_2 + OH^-$. This could explain the fact of lower concentration of $Mg^{2+}$ in our samples as compared to $K^+$. The high $K^+$ concentrations in our samples might be attributed to hydrothermal alteration processes
associated with the high-temperature HTHH eruption. $K^+$ is known to be mobile during hydrothermal alteration and can be leached from the oceanic crust under high-temperature conditions. Studies have demonstrated that altered basalts often exhibit higher $K^+$ content compared to fresh mid-ocean ridge basalt (MORB), suggesting a net flux of $K^+$ into the rock. This increased $K^+$ availability could have contributed to the elevated levels observed in our samples (Millero, F. J. et al., 2007).

Our analysis also revealed a strong 1:1 correlation between $NH_4^+$ and $Cl^-$ ions in the samples collected within the HTHH plume, strongly suggesting the presence of $NH_4Cl$ (salammoniac). This compound forms in volcanic environments due to gas-rock interactions (Martínez-Martínez, J. et al., 2021). The elevated levels of $NH_4Cl$ detected in our samples strongly suggest that the volcanic system could be a primary source of ammonium in the plume. While $Cl^-$ could potentially originate from NaCl, the presence of $NH_4^+$ and $Ca^{2+}$ in our samples suggests a more complex scenario. The observed $Cl^-$ likely stems from reactions
involving $NH_3$ and volcanic gases, such as:

$$NH_3 + HCl \rightarrow NH_4Cl$$
$$2NH_3 + H_2SO_4 \rightarrow (NH_4)_2SO_4$$
$$(NH_4)_2SO_4 + 2HCl \rightarrow 2NH_4Cl + H_2SO_4$$

Additionally, $Ca(OH)_2$ from seawater can react with volcanic gases to produce $(NH_4)_2SO_4$ and CaSO4, further contributing to
$NH_4Cl$ formation: $CaCO_3 + H_2SO_4 \rightarrow (NH_4)_2SO_4 + CaSO_4$ $(NH_4)_2SO_4 + HCl \rightarrow NH_4Cl + H_2SO_4$

$NH_4Cl$, a weak base, dissociates into $NH_4^+$ and $OH^-$ under unfavorable pH conditions. A 5Tg of Cl in the form of NaCl was supposedly injected into the stratosphere after the HTHH eruption with only 0.02% that may have persisted as active chlorine (Zhu et al., 2023). The presence of chloride in our volcanic plume samples highlights the potential for both tropospheric and



stratospheric perchlorate formation, as chloride can participate in various atmospheric reactions contributing to perchlorate production, although the specific limiting factors remain unclear (Jiang et al., 2020; Vömel et al., 2022; Murphy et al., 2024). The HTHH plume's complex chemistry emphasizes the need to consider multiple sources and reactions when assessing ammonium emissions from volcanic eruptions.

### Geographical distribution of cation/chlorinity ratios[1]

| | Na(g/k) Cl(‰) | K(g/kg) Cl(‰) | Mg(g/kg) Cl(‰) | Ca(g/kg) Cl(‰) | Sr(mg/kg) Cl(‰) |
|---|---|---|---|---|---|
| Atlantic | 0.5567 (53) | 0.0206 (44) | 0.06668 (37) | 0.02123 (38) | 0.421 (20) |
| Pacific | 0.5568 (24) | 0.0206 (22) | 0.06671 (24) | 0.02126 (24) | 0.424 (14) |
| Indian | 0.5569 (8) | 0.0206 (8) | 0.06661 (6) | 0.02122 (6) | 0.429 (7) |
| Northern seas | 0.5566 (7) | 0.0205 (6) | 0.06669 (5) | 0.02123 (6) | 0.411 (3) |
| Southern seas | 0.5561 (1) | 0.0207 (1) | — | 0.02126 (1) | 0.423 (1) |
| Mediterranean | 0.5568 (6) | 0.0206 (6) | 0.06668 (7) | 0.02123 (7) | 0.440 (6) |

[1] Number of samples in parenthesis.  Riley, J.P., and Tongudai, M.(1967)

| Ocean etc. | Na (g/kg) Cl ‰ | K (g/kg) Cl ‰ | Mg (g/kg) Cl ‰ | Ca (g/kg) Cl ‰ | Sr (mg/kg) Cl ‰ |
|---|---|---|---|---|---|
| N. Pacific Ocean | 0·5556 (5) | 0·0206 (8) | 0·06670 (10) | 0·02128 (10) | 0·40 (10) |
| S. Pacific Ocean | 0·5554 (6) | 0·0206 (6) | 0·06691 (8) | 0·02128 (8) | 0·40 (6) |
| N. Atlantic Ocean | 0·5552 (7) | 0·0206 (7) | 0·06691 (9) | 0·02128 (9) | 0·40 (6) |
| S. Atlantic Ocean | — | — | 0·06692 (1) | 0·02120 (1) | 0·38 (1) |
| Northern Seas | 0·5553 (5) | 0·0205 (5) | 0·06690 (7) | 0·02121 (7) | 0·39 (6) |
| Southern Ocean | 0·5567 (2) | 0·0206 (2) | 0·06691 (3) | 0·02130 (3) | 0·40 (3) |
| Indian Ocean | 0·5554 (6) | 0·0207 (6) | 0·06696 (10) | 0·02124 (10) | 0·40 (10) |
| Mediterranean Sea | 0·5557 (11) | 0·0206 (11) | 0·06685 (11) | 0·02131 (11) | 0·39 (9) |
| Red Sea | 0·5563 (3) | 0·0206 (3) | 0·06685 (3) | 0·02115 (3) | 0·38 (3) |
| Persian Gulf | 0·5557 (1) | 0·0208 (1) | 0·06695 (1) | 0·02123 (1) | 0·38 (1) |
| North Sea | 0·5541 (2) | 0·0206 (2) | 0·06703 (2) | 0·02118 (2) | 0·40 (2) |
| Baltic Sea | 0·5554 (1) | 0·0205 (1) | 0·06694 (1) | 0·02127 (1) | 0·38 (1) |
| All | 0·5555 (49) | 0·0206 (54) | 0·06692 (66) | 0·02126 (66) | 0·40 (58) |
| Standard Sea Water (Batch P33) | 0·5562 | 0·0205 | 0·06690 | 0·02122 | 0·39 |

(Number of samples in brackets) Culkin, E., and Cox, R.A.(1966)


### Cation/Chlorinity ratios of BraVo aerosol samples analyzed using IC

| Sampler | Cl (µg/m³) | Na (µg/m³) | K (µg/m³) | Mg (µg/m³) | Ca (µg/m³) | SO₄ (µg/m³) | Na/Cl (µg/m³) | K/Cl (µg/m³) | Mg/Cl (µg/m³) | Ca/Cl (µg/m³) |
|---|---|---|---|---|---|---|---|---|---|---|
| Pos# 2 | 1.41 | 1.01 | 0.07 | 0.07 | 1.48 | 0.5 | 0.72 | 0.05 | 0.05 | 1.05 |
| Pos #3 | - | - | - | 0.01 | 0.49 | 0.01 | - | - | - | - |
| Pos# 4 | 1.94 | 1.18 | 0.23 | 0.06 | 1.71 | 0.2 | 0.62 | 0.12 | 0.03 | 0.88 |

**Table.5 Geographical distribution of cation/chlorinity ratios in the ocean as shown by previous researchers versus that observed in the BraVo samples.**










**Average Ionic Constituent Ratios from BraVo Samples (IC)**

| Sampler | Cl⁻/Na⁺ | K⁺/Na⁺ | Ca⁺⁺/Na⁺ | SO4⁻/Cl⁻ |
|---------|---------|--------|----------|----------|
| Pos# 2 | 1.3 | 0.07 | 1.05 | 2.1 |
| Pos# 3 | - | - | - | - |
| Pos# 4 | 1.6 | 0.2 | 0.88 | 0.1 |

**Table. 6 Average ratios of major constituents in seawater reported by Sverdrup et al., (1946), & Junge, C. (1958) versus that**
**observed in the BraVo samples.**

A comprehensive analysis of cation/chlorinity ratios, presented in Tables 5 and 6, reveals significant deviations between our BraVo aerosol samples, collected at stratospheric altitudes (18-20 km), and established oceanic ratios reported by multiple studies (Riley and Tongudai, 1967; Culkin and Cox, 1966; Sverdrup et al., 1942; Junge, 1958). Specifically, the Na/Cl, K/Cl, Mg/Cl, Ca/Cl, and SO₄/Cl ratios in our samples consistently diverge from typical seawater composition. Notably, the Ca/Na

ratios are significantly elevated in our BraVo samples compared to both the Sverdrup et al. (1942) and Junge (1958) data, while the SO₄/Cl ratios show substantial variations. These discrepancies, particularly the elevated Ca/Na and SO₄/Cl ratios in some BraVo samples, strongly suggest potential influences beyond standard marine aerosol composition. Given the timing of our BraVo campaign, which followed 8 months after the HTHH volcanic eruption, these deviations strongly indicate the potential injection of sea salts into the stratosphere by the HTHH eruption. The observed differences in ionic ratios compared

to typical seawater could be attributed to fractionation processes during the volcanic ejection, atmospheric transport, or chemical reactions within the plume. The presence of sea salt in the stratospheric plume is further supported by our analysis of POPC measurements. Therefore, the combined evidence from our BraVo samples and the comparison with established oceanic ratios provides compelling support for the hypothesis that the Hunga Tonga-Hunga Ha'apai eruption injected significant amounts of sea salts directly into the stratosphere.

**6.2 Sulfate fate in the HTHH plume**

We conducted sensitivity simulations using GEOS-Chem, a leading global 3-D chemistry-transport model that incorporates coupled ozone-NOx-VOC-aerosol chemistry for both the troposphere and stratosphere (Eastham, S. D. et al., 2014; Park, R. J. et al., 2004). 0.42 Tg of SO2 was injected at 175° W, 20°S in the model on 15th August 2022 and was equally distributed




between 18.7 km and 28 km. The results (Figure 12) of the run indicate that the balloon would have sampled near 2-6.5 µg/m3

STP of sulfate at the bottom of the HTHH plume but the sampler data indicates below 0.5 µg/m3.

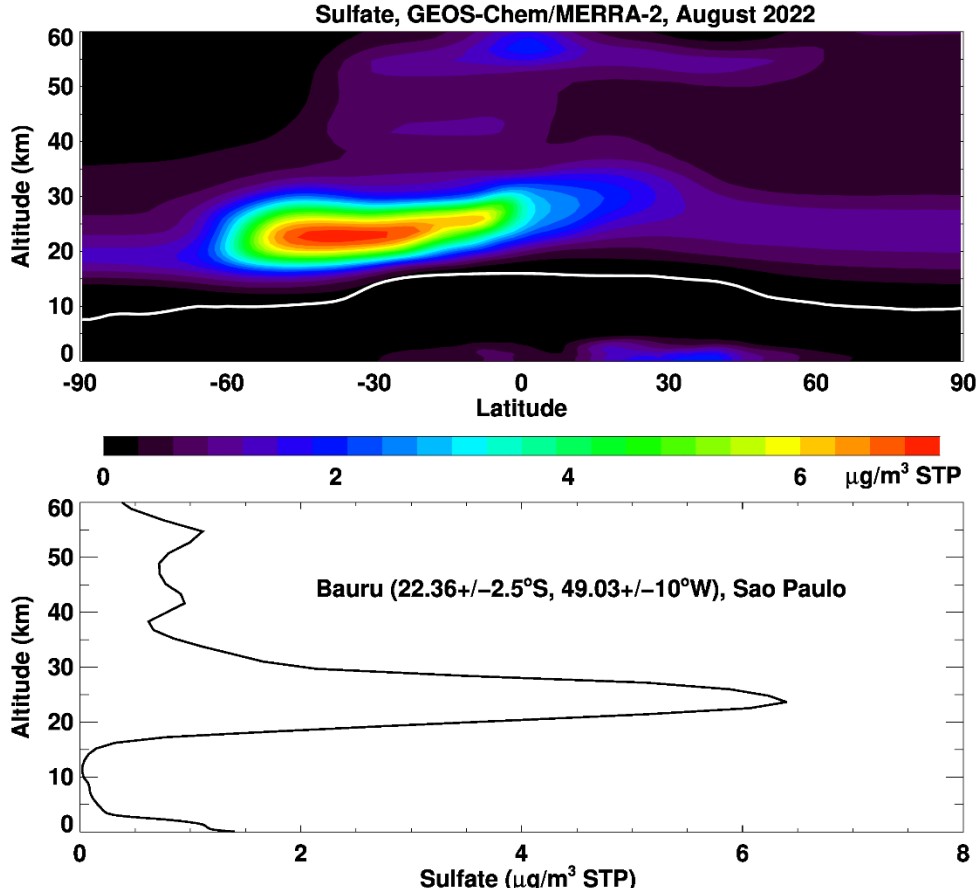

**Figure. 12 Upper panel: Latitude-height cross-section of monthly zonal mean sulfate mass concentrations for August 2022, as simulated by GEOS-Chem. 0.42Tg SO2 emissions from HTHH are injected to 19-28 km altitudes on Jan. 15, 2022, in the model. The white line denotes the location of MERRA-2 thermal tropopause. STP: standard temperature (25C) and pressure (1013.25 hPa).**

**Lower panel: Model simulated monthly mean vertical profile of sulfate mass concentrations averaged within 2.5 deg latitude and 10 deg longitude of Bauru (22.36S, 49.03W), Sao Paulo, for August 2022.**

Satellite-based observations from the Atmospheric Composition Experiment-Fourier Transform Spectrometer (ACE-FTS) indicate the presence of sulfate aerosols in the HTHH plume. The nearest collocated profile, on August 15th near 24.3°S, 45°W, confirms this observation (Figure 13). The weight % ratio of H2SO4/H20 near 70% between 19-22 km and the large

absorption dip near 1200 cm-1 confirmed the presence of sulfate ions (Boone et al., 2022; Bernath et al., 2023) but those relative measurements do not allow the quantity of sulfate concentration. We investigated ACE-FTS from the time of the eruption to the BraVo campaign and found that the spectral signature was largely consistent with the presence of sulfate. However, the spectra over the first 2 months indicate that the dissolution of molecular $H_2SO_4$ into sulfate ions ($SO_4^{2-}$) appears



to be slower than expected. Among other possibilities, the presence of NaCl could explain the lower dissociation rate (Eastes, J. W. (1989) but given the absence of spectral signature within the spectral windows of ACE-FTS, we cannot confirm the presence of NaCl. However, the early presence of NaCl in the plume would also be corroborated by the analysis of the samples collected after the eruption (Colombier et al., 2023) and explain the plume's early ozone depletion (Zhu et al., 2022; Evan et al., 2023).


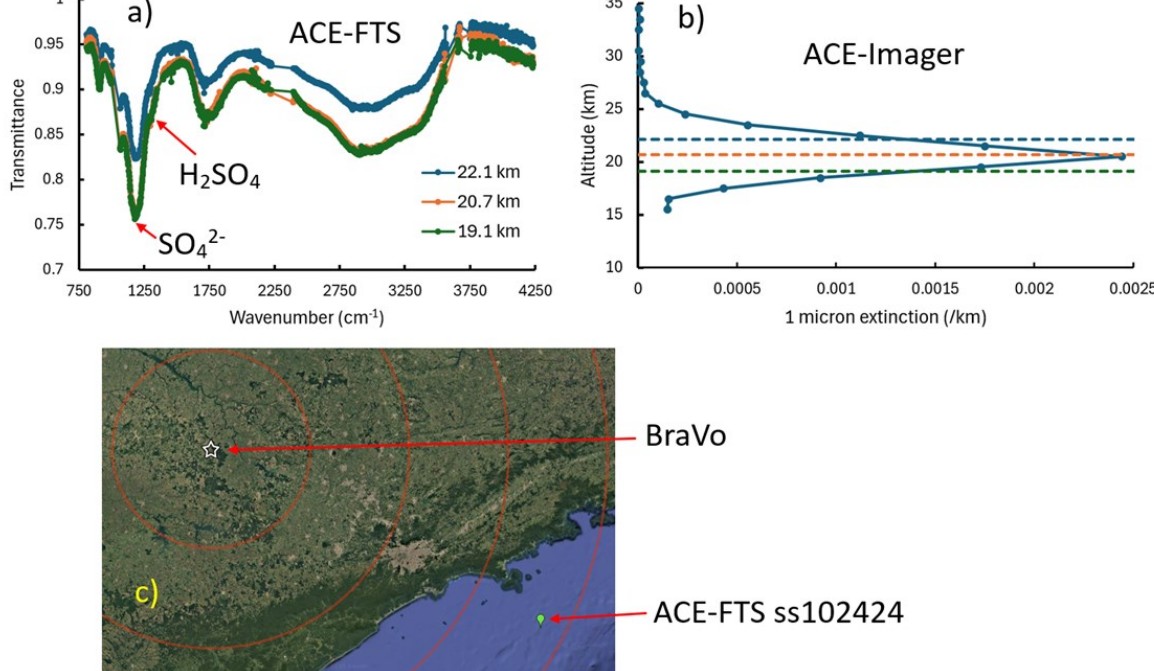

.

**Figure 13.a**) ACE-FTS spectra at 3 altitudes ( 19.1 km, 20.7 km, and 22.1 km) from sunset event 102424 on 15 August 2022, measured
about 6 months after the Hunga eruption. b) corresponding extinction profile at 1 μm from Maestro solar imager instrument quasi-collocated with ACE. c) Position of the measurements (ss102424) relative to the BraVo campaign is shown on the map. Aerial image copyright 2025 © Google, TerraMetrics.  Orange circles on the figure are in increments of 125 km in radius, to provide scale.  The occultation is 465 km away from the balloon site.

The low sulfate concentrations observed in our IC analysis could be due to interactions of $SO_4$ with $Ca^{2+}$ forming $CaSO_4$ and other components during the extraction process, potentially leading to sulfate loss or transformation. Studies have shown that $CaSO_4$ can form hy drated calcium sulfate compounds, leading to the precipitation of gypsum and the potential loss of sulfate



(Van Driessche, A. E. S.et al.,2019) before entering the IC. Additionally, the presence of NaCl in the plume could have further influenced sulfate concentrations. Deliquescent salts like NaCl can absorb water vapor, creating a humid microenvironment

that can promote the dissolution of $CaSO_4$ and subsequent gypsum formation (Pierre Bracconi et al.,2020). Our sampling and analysis methods might have been unable to detect low concentrations of sulfate aerosols, especially if they were present or split into smaller particles that would have eventually been lost during filtration.

Water vapor concentration from a CFH radiosonde, launched on August 12th during a SAGE III/ISS overpass, was measured and compared with SAGE III/ISS and MLS profiles. The results showed an enhanced water vapor concentration peaking near

25 km, associated with the Hunga Tonga-Hunga Ha'apai (HTHH) volcanic plume (not shown).

This significant water vapor injection plays a crucial role in understanding the observed low sulfate concentrations in our analysis. An accurate volcanic $SO_2$ lifetime is essential for predicting aerosol particle sizes within volcanic clouds. Shorter $SO_2$ lifetimes result in higher aerosol concentrations, leading to more rapid coagulation and the formation of larger sulfate particles (Clyne, M. et al., 2021). In the stratosphere, the $SO_2$ lifetime is primarily governed by its reaction rate with OH

radicals and heterogeneous reactions on ash particles (Zhu et al., 2020). Typically, in $SO_2$-rich plumes, OH depletion slows the $SO_2$/OH reaction, prolonging the $SO_2$ lifetime (Mills et al., 2017; Zhu et al., 2020; Bekki, S. 1995; Pinto et al., 1989; Savarino et al., 2003). However, the HTHH eruption stands out due to the substantial amount of water vapor injected into the stratosphere. This water vapor led to a rapid increase in OH radical concentrations (Zhu et al., 2022), effectively shortening the $SO_2$ lifetime.

The observed water vapor enhancement near 25 km directly supports the hypothesis of increased OH concentrations. This, in turn, suggests a shorter $SO_2$ lifetime and, consequently, the potential for larger sulfate particle formation through enhanced coagulation. This process likely contributed to the lower sulfate concentrations observed in our analysis, as larger particles have a greater tendency to sediment out of the stratosphere.

| SAMPLES | Cl/Na ratio | SO4/Ca ratio | Cl/K ratio |
|---|---|---|---|
| Col_23_HT9 | 1.17189038 | 1.10063369 | 101.1048425 |
| Col_23_HT1 | 1.19594678 | 1.053711349 | 78.92392975 |
| Col_23_HT6 | 1.203960857 | 1.109184673 | 41.53413081 |
| **BraVo_Pos# 4** | **1.066673042** | **0.036526878** | **3.29191711** |
| Volkilau_0612 | 1.285031735 | 6.286580147 | 48.21084578 |
| Volkilau_0614 | 1.151113052 | 35.45939526 | |

**Table.7 Ratios of Cl/Na, SO₄/Ca, and Cl/K were calculated using ion mixing ratios (in mol) from three sources: (1) ground samples collected near Hunga volcano within the first two weeks after the eruption (Colombier et al., 2023); (2) results presented in this study (BraVo-pos#4); and (3) samples collected in Hawaii after the Kilauea eruption (Vernier et al., 2018).**



We compare our stratospheric plume sample results with ground samples collected within two weeks of the Hunga eruption
(Colombier et al., 2023) and with ground samples collected ~60 km downwind of the Kilauea eruption in Hawaii using a 3-
stage aerosol impactor (Vernier et al., 2018) (Table 7). While sampling conditions varied considerably, these comparisons aid
in the interpretation of our key findings.

The presence of sea salt is evident in all samples, as indicated by Cl/Na ratios near unity (ranging from 1.06 for BraVo-pos#4
to 1.28 for VolKilau_0612). In the Hawaiian airborne samples, sea salt was found on stage 1 of the impactor, confirming the
presence of large particles (r > 1.4 µm). The Ca/SO$_4$ ratio in the ground samples collected by Colombier et al. (2023), which
is near unity (Table 1, Col23_HTx) and indicative of sea salt influence, differed significantly from the ratios observed in the
stratospheric and Hawaiian samples. In the Hawaiian samples, sulfate was partially neutralized by ammonium (NH$_4$), forming
ammonium sulfate. However, the large excess of NH$_4$ relative to SO$_4$ in the stratospheric samples suggests that this
neutralization process did not occur. These findings imply that gypsum was present in the ground samples. The formation of
gypsum can influence the availability of sulfate for other aerosol species, including ammonium sulfate (Miyamoto et al., 2019).
Since gypsum was not directly emitted by the degassing eruption in Hawaii, it likely did not persist in the stratosphere. We
note here that Ammonium nitrate was found on stage 3 of the impactor in Hawaii implying that the particles were below
r<0.2µm. A significant fraction of such particles would be lost if filtered by the aerosol sampler used in this study with pore
size near 0.45µm. This may explain the very low sulfate concentration found on the BraVo-Pos#4. Finally, Cl/K was always
very large in Hawaii again reflecting the importance of sea salt relative to other combustion sources. While there is a very
large excess of Cl (ratio>40) for Hunga ground samples relative to Hawaii, this result is very different for the stratospheric
samples showing only a factor 4 difference. Overall, the key findings of the table presented in here are:

- Na and Cl found in the stratospheric Hunga plume are remarkably similar to what would be expected in sea salt but other
elements do not appear consistent such as the large amount of K or NH$_3$.

- Ca and SO$_4$ during BraVo are very different than in Hunga ground samples and thus likely excluding the presence of
gypsum that could have survived stratospheric conditions probably consistent with the removal process (scavenging and
sedimentation)

- The low levels of sulfate in the stratospheric samples could be explained by the filtering efficiency of our system retaining
large particles while sulfate was likely smaller than the pore size (0.45 µm) of the filters used for aerosol collection using the
lightweight aerosol sampler.

The result presented here appears to contradict the idea that the Hunga plume was largely composed of sulfate aerosol more
than 6 months after the eruption. An example of spectra from the ACE-FTS instrument, taken within 500 km and 3 days after
our balloon flight, strongly indicates the presence of sulfate aerosol based on distinct absorption features (Fig.13). This
apparent contradiction may stem from several factors. Firstly, ACE-FTS, as a remote sensing instrument, detects the integrated
aerosol burden, including particles of various sizes. Our balloon-borne sampling, utilizing filters, may have preferentially
retained larger particles while potentially losing smaller sulfate-rich particles below the filter's pore size (0.45 µm). Secondly,
the eight-month time difference between the eruption and our measurements is significant. Stratospheric aerosols undergo





substantial evolution through processes like coagulation, sedimentation, and chemical reactions. Thus, while ACE-FTS captured the plume's composition shortly after our flight, the aerosol population could have undergone significant changes

leading to the observed differences in our in-situ samples.

Our balloon-borne measurements of the Hunga Tonga volcanic plume revealed an unusual chemical composition, characterized by elevated calcium concentrations and a relative sulfate depletion. This observation contrasts with typical marine aerosol compositions, with more consistent sulfate and chloride ratios. To explain the observed sulfate deficit, we propose that gypsum ($CaSO_4 \cdot 2H_2O$) formation acted as a significant sink for sulfate within the stratospheric plume. This

hypothesis is supported by recent experimental work by Kellermeier et al. (2024), who demonstrated the nucleation and growth of gypsum under controlled conditions relevant to the stratospheric environment. Their titration assay, simulating the gradual increase of calcium and sulfate ions at $p^H$ 8.0, showed that gypsum precipitation occurs readily under conditions of supersaturation, leading to a reduction in dissolved sulfate. The high calcium concentrations observed in our Hunga Tonga plume samples, combined with the presence of sulfate, would have created conditions conducive to gypsum formation.

Consequently, the rapid nucleation of gypsum within the plume would have effectively removed sulfate from the aerosol phase, resulting in the observed low sulfate-to-chloride ratios. This finding underscores the importance of considering mineral formation processes in understanding the chemical evolution of volcanic plumes and their impact on stratospheric aerosol composition.

The lack of sulfate in our samples is likely real but also highlights a potential deficiency in our method for collecting small

aerosols using filters. The pore size of a filter is a tradeoff between the strength of the pump and flow. Alternative sampling methods with impactors could improve collection efficiency, but their weight limits balloon deployment. Despite these limitations, the unique results presented here offer valuable insights.

## 7. Conclusions

In this study, we investigated the chemical composition and evolution of the stratospheric plume eight months after the unprecedented Hunga Tonga-Hunga Ha'apai (HungaTHH) submarine volcanic eruption of January 2022. This eruption injected an exceptional mass of seawater directly into the stratosphere, providing a unique opportunity to examine the impact of such events on stratospheric aerosols. Utilizing balloon-borne data from the BraVo campaign in Brazil, we analyzed the optical, microphysical, properties and chemical composition of the plume.


Our analysis revealed a chemical composition that significantly deviated from typical sulfate aerosols, suggesting complex transformations and multiple source contributions. The presence of sea salt-like aerosols, indicated by elevated $Na^+$ and $Cl^-$ levels, confirmed the direct injection of marine aerosols into the stratosphere. However, discrepancies in other ionic ratios, particularly the elevated Ca:Na and $SO_4$:Cl ratios, pointed to additional sources and chemical transformations beyond standard

marine aerosol composition. The significantly high Ca:Cl ratio, exceeding seawater values, suggests the influence of volcanic material weathering, hydrothermal alteration, and riverine inputs, highlighting the intricate interplay between volcanic

emissions and the marine environment. The detection of NH₄Cl (salammoniac) and the strong 1:1 correlation between $NH_4^+$ and $Cl^-$ ions further underscore the complex gas-rock interactions within the plume. The formation of NH₄Cl, coupled with the potential for both tropospheric and stratospheric perchlorate formation, resulting from the presence of biomass burning and

NH₄ indicates the plume's significant influence on atmospheric nitrogen chemistry and potential implications for ozone depletion.

The observed sulfate deficit in our samples, despite satellite observations indicating sulfate presence, presents a compelling case for the role of gypsum ($CaSO_4 \cdot 2H_2O$) formation as a significant sulfate sink. This hypothesis, supported by recent

experimental studies, highlights the importance of considering mineral formation processes in understanding the chemical evolution of volcanic plumes. Furthermore, the potential loss of sulfate particles due to our filtration technique's pore size and the eight-month time difference between the eruption and our measurements likely contributed to the discrepancies between our in-situ measurements and satellite observations. The rapid nucleation of gypsum, driven by high calcium concentrations, likely led to the effective removal of sulfate from the aerosol phase, during our sampling and analysis, explaining the observed

low sulfate-to-chloride ratios.

The discrepancies between our in-situ measurements and satellite observations underscore the challenges of accurately characterizing stratospheric aerosols. Our balloon-borne sampling, utilizing filters, may have preferentially retained larger particles while potentially losing smaller sulfate-rich particles. The eight-month time difference between the eruption and our

measurements further emphasizes the dynamic evolution of stratospheric aerosols through coagulation, sedimentation, and chemical reactions.

The HungaTHH eruption's unique characteristics, particularly the substantial water vapor injection, significantly altered the stratospheric environment. The increased water vapor likely led to elevated OH radical concentrations, shortening the SO₂

lifetime and potentially enhancing sulfate particle formation through coagulation. This process likely contributed to the lower sulfate concentrations observed in our analysis, as larger particles have a greater tendency to sediment out of the stratosphere.

Comparison of our findings with ground samples collected shortly after the eruption and with samples from the Kilauea eruption provides valuable insights into the evolution of volcanic aerosols. The presence of sea salt in all samples, indicated

by Cl/Na ratios near unity, highlights the ubiquitous nature of marine aerosol influence. However, the differences in Ca/SO₄ ratios and the absence of gypsum in the stratospheric samples, compared to ground samples, underscore the dynamic chemical transformations occurring during atmospheric transport.

We compared collocated balloon measurements (POPC, COBALD) with SAGE III/ISS, using refractive index properties of

sulfate and sea salt to derive extinction coefficients from these in situ measurements. We demonstrate that the similar refractive





index properties of sulfate and sea salt in the UV-Visible range do not allow us to conclusively determine the presence of sea salt in the plume.

In conclusion, our study demonstrates the complex chemical composition and dynamic evolution of the HungaTHH volcanic plume, highlighting its potential impacts on atmospheric and climate understanding. The direct injection of sea salts into the stratosphere, coupled with the formation of gypsum and the influence of water vapor on sulfate concentrations, underscores the need to consider a multitude of factors when assessing the impact of submarine volcanic eruptions on the stratosphere. Our findings also highlight the limitations of current sampling techniques and the need for further research to accurately characterize the complex aerosol environment within volcanic plumes. This research contributes valuable insights into the

long-term impacts of such events on global climate and atmospheric chemistry, emphasizing the importance of continued monitoring and modeling of volcanic emissions.

## Data availability

All raw data can be provided by the corresponding authors upon request.


## Author contributions

DQ, HV, EL, and JPV planned the campaign; HV, DQ, BB, GL, FL, AS, AM, JM, and JPV performed the measurements; HV, DQ, FL, GS, DP, BG, HL, AR, MK, FW, CB, and JPV analyzed the data; HV and JPV wrote the manuscript draft; DQ, BB, EL, FL, NR, HL, SF, FW, ND, MC, GB, MJ, MA, JPV, and ND reviewed and edited the manuscript.


## Acknowledgments

This research was funded by the NASA Upper Atmospheric Observation Composition (UACO) program and the Atmospheric Composition Modeling and Analysis Program (ACMAP).

Additional support was provided by the Agence Nationale de la Recherche (ANR) through grant ANR-LABX-100-01 from

Labex VOLTAIRE, managed by the University of Orleans.

We thank Amit K. Pandit and Johnny Mau for their invaluable technical assistance in preparing the balloon campaign. We are grateful to Yunqian Zhu and Xinyue Wang for their insightful discussions and contributions to WACCM simulations.

## Competing interests

At least one of the (co-)authors is a member of the editorial board of Atmospheric Chemistry and Physics.



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
