# Peer review of "Balloon Observations Suggesting Sea Salt Injection into the Stratosphere from Hunga Tonga-Hunga Ha'apai"

_EGUsphere, 2025_

## Referee Comment (RC1)

**Review of the ACP manuscript acp-2025-924**

"Balloon Observations Suggesting Sea Salt Injection into the Stratosphere
 from Hunga Tonga-Hunga Ha'apai"

By Hazel Vernier, Demilson Quintão, Bruno Biazon, Eduardo Landulfo, Giovanni Souza, V. Amanda Santos, J. S. Fabio Lopes, C.P. Alex Mendes, A.S. José da Matta, K. Pinheiro Damaris, Benoit Grosslin, P.M. P. Maria Jorge, Maria de Fátima Andrade, Neeraj Rastogi, Akhil Raj11, Hongyu Liu, Mahesh Kovilakam, Suvarna Fadnavis, Frank G. Wienhold, Mathieu Colombier, D. Chris Boone, Gwenael Berthet, Nicolas Dumelie, Lilian Joly, and Jean-Paul Vernier, 2025

The above manuscript describes a detailed analysis of measurement data derived for a quite unusual volcanic eruption. The authors use satellite, lidar and in situ balloon-borne measurement data for this analysis with a focus on optical and chemical plume particle properties. Even if the number of balloon flights is relatively small, the lack of in situ data points of this event make every single data point valuable. Consequently, the manuscript should be published, although I believe it needs some polishing and restructuring beforehand. The text seems rashly written (there are many mistakes) and the structure needs some improvement and streamlining. For instance information about the scientific background is given at several places in the manuscript (and not restricted to the introduction) and also instruments characteristics are described in different sections. Section headlines are not consistent and it is hard to follow the line of arguments, because it is not obvious, what will be discussed next. Additionally, the manuscript is quite lengthy and should be shortened to focus more on the main messages.

Even if this eruption would be the only event of its kind in a century, it is an interesting real-world experiment made by nature. Nevertheless, I miss the information about how frequently such events are expected in the introduction. As an atmospheric researcher I do not know this, but a volcanologist might be able to give an estimate or an upper limit.

**Specific remarks:**

p. 1, l. 32: Even if the measurements have an uncertainty, the SAOD was either lower compared to previous eruptions or it was not, but "appears" seems to me the wrong term.

p. 2., l. 64: "Paradigm" is a big word. This would fit if the occurrence of such eruptions is frequent (s. a.). "View" is probably the better term in this context.

p. 2., l. 77: Analyzing the plume is relevant, but what is to goal in doing this? Providing a better understanding or even a parametrization to the modelers? Understanding satellite data? It is not clear where the manuscript is heading.

p. 5., l. 125/126: What about the Sao Paulo measurements? Do they look similar?

Three kilometers difference in the top of the aerosol layer seems to be much, are there any other measurements available which support the zonal averaging theory?

Please explain more in detail the argument of no CALIPSO measurements over Brazil due to SAA, for a non-specialist this is not automatically clear.

p. 11., l. 249: This paragraph should be part of the introduction, but does not belong here.

p. 12., l. 271: The sentence containing "… both eruptions …" seems to be copied from somewhere as in Fig. 6 more than two eruptions are shown. This does not fit.

p. 15., Discussion on level Pos #2: It is not clear to me, why the discussion on these low level data is needed. This holds as well for Fig. 8. For the main topic of this manuscript, it is not needed. The only reason could be to validate the sample data versus ground station data to show that the sampler and the post-flight analysis works well. Otherwise this discussion should be deleted.

p. 17., Discussion on Fig. 9 and figure caption: The measurements were made 7-8 months after the eruption. Hence one would assume an already well mixed stratospheric aerosol. But the lower row of plots in Fig. 9 shows something different. What is this reason for this, for me unexpected and still large variability? And is this confirmed by, e.g. model results?

p. 18., l. 376: This paragraph is instrument characteristics description and does not belong here but should be moved to an earlier section.

p. 19., last paragraph: The discussion about the extinction profiles and also Fig. 10. As we are talking about a volcanic eruption can you rule out that there has not been any absorbing material in the stratosphere? How would the theoretical response functions look like if you assume e.g. 0.5 or 1% BC? And how would the comparison of the extinction in Fig. 11 be affected due to this?

p. 26., l. 564: "Standard temperature" is 273 K, hence 0°C, 20°C or 25 °C is "normal temperature", right?

p. 25., l. 555, the discussion about the sulfate concentration: The particles are sampled at -60°C and then experience +25°C in the sampler for at least 30 min, according to Fig. 5. How does this influence your sampled sulfate, there must be some evaporation?

p. 28., l 615 Tab. 7.: Please delete most of the digits in the numbers, three leading digits should be enough. Please provide the information on the campaign in a new column, as by the upper measure you saved space. What does "Volkilau" mean?

p. 29., l. 639: What does "sea salt relative to other combustion sources" mean?
Sea salt is not from a combustion source. And combustion sources have not been discussed beforehand.

p. 29., l. 646, The whole paragraph: The difference between the different sulfate measurements is discussed here once again. Please merge all sections with this topic and make it only one section, the reader gets lost otherwise.

p. 31., l. 689: The "presence of biomass burning" is mentioned for the first time in the conclusions. But this should not be the case, either biomass burning plays a role, then it should show-up already in the former sections, or it should be deleted in the conclusions.

p. 31., l. 708: The discussion in this paragraph, I again miss "evaporation" as potential reason.

p. 31., l. 710: You probably mean "sulfate particle mass formation" but not new particle formation.

**Technical corrections:**

p. 1, l. 46: The sentence reads strange, I assume that a "which" and a comma is missing before "are known …", please check.

p. 1, l. 46: Referring to table 1 is misleading, as there is only one major (VEI = 6) volcanic eruption listed. I suggest to delete this reference here.

p. 1, l. 48: A paper reference to the statement about the 0.42 Tg $SO_2$ is missing.

p. 4, l. 102: "Therefore, …" is not correct, the above statement is not the reason, why HTHH did not produce a significantly larger SAOD (this is due to the kind of eruption), but why they look similar in the Fig. 1.

p. 5, l. 114: I did not know the city of Bauru, hence I looked at google maps and for me the city seems to be only 250 km west to Sao Paulo. Are you sure about the 400 km? Or is it 400 km distance pointing northwest?

p. 5, l. 126: Please start a new paragraph after "(SAA)." as the next sentence is about a different topic.

p. 6, l. 138: Please start the headline of section 3.1 with "Balloon-born instrument …"

p. 6, l. 140: A space is missing in "2 kg"

p. 6, l. 145: Any reference paper out on the POPC which could be cited in this paragraph?

p. 8, Fig. 4: The print quality of this figure is poor (pixels visible), please provide a better version.

p. 8, l. 190/193: Please write out the acronyms ECC and CFH when you use them the first time.

p. 9, l. 204: The first paragraph can be deleted as this information was already given on page 7 and is repeated later once again.

p. 10, l. 237: please exchange ", &" with "and".

p. 10, l. 238: is the given ratio "12.8 ng/6ml" correct or is it "12.8 ng/ml"? in the former case a space would be missing.

p. 10, Fig. 5: As you write about the detection range of the POPC in Sec. 3.1.1 in terms of radii, it would be good to add at the legend here "particle diameter", at least that is what I assume.

p. 11, l. 242: Please delete one of the two "((".

p. 11, l. 242: The sentence "The analysis involved …" is a repetition of the short sentence in line 241.

p. 12, Fig. 6: Please add text to the color scale (what is displayed), otherwise the figure is hard to understand.

On the y-axis is SAOD or sSAOD displayed, as stated in the text?

p. 12, l. 272: The "Since" must be deleted, otherwise a half-sentence is missing.

p. 13, l. 282: I´m not a native speaker, but I believe "non-light" is not correct, what should this be, darkness? You mean not light absorbing, please check and rephrase.

p. 14, l. 329: Please delete the comma after "emissions".

p. 16, l. 346: "AMSL" is not explained.

p. 17., l. 363: The reference to Fig. 10 seems to wrong, should be Fig. 11, as Fig. 10 shows something different (as correctly described on page 18).

p. 18, l. 382: Please exchange "H.(2022)" with "H., 2022)"

p. 18, l. 384: The "size of the POPC" hopefully stays constant, what you mean is the "particle size derived by the POPC".

p. 19, l. 395: The unit for an hour should be "h" not "hr", please exchange.

p. 19., l. 396: The name of the instrument "Compact Optical …" should be stated in Sec. 3.1.2, but not anymore in the results section.

p. 19, l. 399: The reference to the "Methods", what is meant with this, an appendix? Or a former section of the manuscript?

p. 21, l. 448: The format of the Adachi and Buseck reference, there a two opening parentheses but only one closing parenthesis. Some of the references in this section use the "and" others the "&". Please harmonize in the whole document.

p. 22, l. 458: Space missing after "eruption".

p. 22, l. 461: The name of the volcano and its abbreviation was already introduced beforehand, hence use only HTHH here. Same line 600. Later in the manuscript many different versions of the name are used "Hunga" samples or plume, "Hunga Tonga" plume, "HungaTHH" etc., please harmonize.

p. 22, Tab. 4: Table caption, should be "Zhang".
And in the first row should be "Pos# 4" and not "Pos3 4".

p. 23, l. 493: In the reaction equation the charges on both sides are not equal, but this should be the case, right?

p. 23, l. 509: The "4" in $CaSO_4$ should be subscribed,. Please check the whole manuscript.

p. 23, l. 511: A space is missing in "5 Tg".

p. 24, l. 520: Table 5 is of poor print quality, partly to small characters and overcrowded with information. How about extracting the essential information from the literature and include it in the newly made lower part of the table?

p. 27, l. 592: The space in "hy drated" must be deleted

p. 29, l. 633: Space missing before the "μm".

---

## Referee Comment (RC2)

Herein the authors present data collected during the BraVo campaign wherein they collected data within the HTHH plume over Brazil. The authors present data collected from balloon-borne instruments, satellite instruments, model, and ground instruments in an attempt to better understand the composition of the particles in the plume. This paper represents a massive amount of work that went into collecting, analyzing, and interpreting the data; I commend the authors for this effort.

I have to be frank, this paper was very frustrating to read. I apologize if my frustration came through in the review, but I cannot believe that any of the 24 co-authors put a serious effort into reading this paper or evaluating the methodology and conclusions prior to its submission. It is very frustrating to, as a reviewer, do the work that should have been done by co-authors. That said, I endeavored to keep the criticism constructive, if blunt at times.

In my view, this paper has a very long road to travel before it is ready for peer review. The overall structure of the paper is very disjointed and confused, it lacks focus, and does not provide sufficient evidence to accept the authors' conclusions. There are many sections that repeat information in previous sections, paragraphs containing instrument/measurement details appear in the main body of the text when they should have been in section 2, there is a lack of detail that precludes me from understanding what methodology the authors employed (much less repeat the experiment myself), there are contradictory statements and conclusions drawn about the data, etc.

Perhaps the most disconcerting aspect of this paper is the number of times the authors admit that their data do not match what is expected, based on previous work, but follow that admission a "just-so" explanation. I apologize for using that phrase, but the authors continually defend their conclusions, despite what the data actually support, by invoking phrases such as "it is possible", "could be due to", "potentially", "might be", etc., all the while failing the provide definitive evidence for their conclusions. For example, to justify the disagreement between their measured ion ratios and those published and accepted in the literature we receive statements such as (all emphases added):

1. line 340: "This suggests *potential* exposure to wildfire smoke...", but their back trajectories do not go over the burn areas.

2. line 343: "it's also *possible* that some air parcels *could have* originated from the Eastern Pacific". Possible, yes, but their back trajectories do not support this.

3. line 471: "...it's *possible* that additional sources contributed to higher concentrations of other ions..." What other sources? Where is the support?

4. lines 482-487: "Ca2+ could *potentially* be associated with gypsum...this ratio *could* arise from two primary mechanisms...the observed ratio alone cannot definitively distinguish between them."

5. lines 590-591: "The low sulfate concentrations observed in our IC analysis *could* be due to interactions of SO4 with Ca2+...*potentially* leading to sulfate loss"

6. lines 610-613: "This, in turn, suggests a shorter SO2 lifetime and, consequently, the *potential* for larger sulfate formation through enhanced coagulation. This process *likely* contributed to the lower sulfate concentrations in our analysis..."

7. lines 650-651: "Our balloon-borne sampling, utilizing filters, *may have* preferentially retained larger particles while *potentially* losing smaller..."

8. line 669: "The lack of sulfate in our samples is *likely* real but also highlights a *potential* deficiency in our method..."

9. lines 698-699: "The rapid nucleation of gypsum, driven by high calcium concentrations, *likely* led to the effective removal..." I note that the authors did not demonstrate the gypsum was formed, it is only a convenient explanation for their low sulfate ratio.

10. line 710: "...*potentially* enhancing sulfate particle formation..."

This is only a representative, not exhaustive, sample from the text, but it demonstrates the point that their observations do not match literature values for sea salt and they have no proof of their alternative explanations.

I do not necessarily disagree with the author's claim that sea water (salt water) persisted in the stratosphere for months. What else could we expect from an injection of 146 Tg of sea water? But I have to ask which is more plausible: The authors' correctly identified every *potential* reason for their data being significantly different from the accepted standards (despite the lack of scientific evidence to support their claims), or their sampling was flawed and their samples were contaminated? I find the latter more likely and I believe all of the evidence, as presented within this paper, points directly to that conclusion. Interestingly, the authors acknowledge this possibility (lines 623/635) as well, but do not embrace this possibility.

Everything in this paper points to either contaminated sampling or some very interesting atmospheric chemistry and the only support we receive from the authors are passing statement of it *could* be something else. If the former, then the conclusions that can be drawn from this work are very narrow and the paper requires a massive rescope and revision. If the latter, then the authors must provide definitive proof and the paper, again, requires a massive rescope and revision. Again, all of the IC data presented in this paper point to contamination. Therefore, I see no opportunity for either of these options playing out. It is for these reasons that I must recommend rejection without further review.

Below are specific comments regarding the text.

1. The abstract reads like an introduction. Please rewrite as an abstract.

2. line 45: I do not know what the sentence that begins with "Unlike major..." means. Please clarify.

3. line 55: is SO4 missing its charge?

4. Section 2: there is no discussion of balloon measurements.

5. line 99: "SAOD" was defined earlier.

6. line 100: Kelvin does not use the degree symbol.

7. line 100: Sentence beginning with "Given that the...", I do not understand what the authors are trying to communicate. Please revise.

8. line103: "...contrary to previous claims." This is an unsubstantiated claim and is made en passant. The authors fail to discuss the differences in calculation and the impact this may have on the results. If the authors wish to make this claim they should do so in the body of the text alongside a thorough discussion of why their method is correct.

9. Figure 1: this figure makes no sense here. If you wish to use it, please move it to an appropriate location.

10. line 114: "400 km west", west of what?

11. Figure 2: please include panel labels. Please make range of y-axes the same for each panel so the reader can better appreciate your visualization. Please make the font size for the ground-based lidar profiles larger (x/y axes). Please make the map's scale larger so it can be read. As is, the map is not useful; please zoom in to better show location in Brazil.

12. line 124: no need to redefine CALIPSO.

13. line 126: This reads better if you create a new paragraph between sentences here.

14. lines 130-135: Text better fits in section 2.

15. "sampler was coupled with a radio-controlled valve attached to the balloon's neck". Why? What did this accomplish? Please clarify why this is important.

16. Section 3.1: All subsections belong under section 2.

17. line 140: Should "OPC" be "POPC" here? If not, please explain the difference.

18. Section 3.1.1: The authors describe 2 versions of the POPC. 1 with 8 channels, the other with 30 channels. Please include the size bins for each (you use this information later in the paper) and please clarify which version you use in the following analyses/figures.

19. line 150: what is meant by "phase-sensitive detection"?

20. line 154: What is meant by scattering ratio? Is this the color ratio?

21. Figure 3: Within the legend, what is "NPOPC"?

22. Figure 3 caption: What is meant by "backscatter"? Is this the returned signal intensity; why is it greater than 1?

23. line 164: Should this be a new section or subsection? Fits better in section 2.

24. line 171: reference to "Fig. 7" is way to early (you just finished discussing Fig.3!) and is not needed anyway. Please revise for clarity or just remove.

25. line 174: Reads better if there is a paragraph break here.

26. line 180: Where in France?

27. Figure 4: Extending the plots to 3.5 hours after the landing does not make sense and seems like a waste of page space. If this is needed, please tell the reader, otherwise eliminate the unneeded portion.

28. line 189: Should this be a new section? Fits better under section 2.

29. line 189: Neither ECC nor CFH are defined in the text. Please define.

30. line 189 ff: Please include manufacturer and model for this equipment.

31. line 192: Please provide a reference for the accuracy claim. There are a plethora of papers on ECC and CFH instrument/measurements. Can the authors cite an appropriate number of them to help curious readers better understand these measurements?

32. line 195: Is it dew point or frost point? They are different; please clarify.

33. line 197: Please provide a reference for the 4% claim.

34. line 199: Can the authors provide a guide for what is meant by "middle stratosphere and tropical tropopause"? What altitude does this specifically mean for your measurement location?

35. Table 2: This table is never reference in the text. Do you need it?

36. Table 2 caption: Why the period in "Table.2"? This occurs in all table captions. Please revise as needed.

37. line 204: What "previous balloon-borne aerosol collection efforts" are you talking about? Whose? When? Where? Please provide context and references.

38. line 207: Is there a reference for the India work?

39. lines 204 - 210: This reads very similar to the earlier "Sampler" section. Please consider condensing.

40. lines 217-218: This information is also redundant with that presented in a previous section. I see no value in restating this (nearly verbatim); please revise or remove.

41. Figure 5: In section 3.1.1 you used radius, here you use diameter. Please be consistent.

42. line 239: It is unclear where the 12.8 ng number came from. Doesn't this assume a total volume of 6 mL? Please explain.

43. line 239: The authors failed to explain what is meant by filter positions #5 and #8. Please provide enough detail to allow the reader to understand (especially those that have not done IC).

44. Section 3.3: Overall, this entire section requires a thorough revision to communicate clearly. As written, I could not reproduce the method and I am unable to understand it well enough to appreciate your work.

45. lines 249-256: Much of this text is out of place and fits better in the introduction. Please revise for clarity and brevity.

46. Figure 6: label the color bar and include units

47. Figure 6 caption: "Colored circles depicting peak volcanic SO2 emissions" Really? Why does this only go to 1? Your table 1 has Pinatubo at 21 Tg. Is this scale relative to Pinatubo? Also, from the text I would have guessed this is the SO2 / SAOD ratio. Please ensure the colorbar label, the caption, and the text tell the same story.

48. Figure 6 caption: ...versus the corresponding semi-annual mean (sSAOD)..." The y-axis and title of Figure 6 show SAOD, not sSAOD. Please ensure consistency.

49. Figure 6 caption and line 274: sSAOD has different definitions in the caption and in the text. Please correct.

50. line 268: GloSSAC already defined.

51. line 278: Why compare only with the largest eruptions? Perhaps it is better to say "When compared with past eruptions of comparable VEI magnitude..."? Event then, you should note that VEI is not an indicator of ejected SO2 mass. VEI is a measure of the volume of ejected tephra, so I am unsure how the sentence in question is relevant to the paper.

52. line 298: This sentence is out of place and should be removed. Perhaps more appropriate in the introduction.

53. line 336: "only $NO_2^-$ was present in reportable quantities at Pos #4. Due to associated error bars, other ions measured at this position were not statistically significant..." I cannot see NO2- on the bar plot of Figure 7. If NO2- is "statistically significant" and NO3 and SO4 are not, then I certainly have no idea how to interpret this figure or your results. Please clarify and correct.

54. line 341: What is meant by DeepBlue AOD? This is put in the text with no introduction. Can you provide something for the reader so he won't have to search (e.g., a brief statement of what this is, a reference, etc.).

55. line 343: "also possible..." This is pure speculation that is not supported in any way by anything you have shown. This is grasping at straws to support a predetermined conclusion. Either support this claim, or remove it from the paper.

56. line 355: In Section 3.1.1 you stated POPC has either 8 or 30 bins; here you say 10. Please clarify and/or correct.

57. line 363: Should be Figure 11.

58. lines 363-368: Here, you discuss extinction before explaining how you calculated extinction. You say the method is explained in the next section, but the next section references a non-existent section! Please ensure your argument is presented in a logical and coherent manner.

59. Figure 9: The text in this figure (axis labels, tick labels, legend text) is very small and the x-axis tick labels are crammed together so much that it is hard to read. Please consider improving the readability of this figure.

60. Section 6: Much of the content in this section should be moved to section 3.1.1.

61. line 384: "...and therefore the size of the POPC within the HTHH plume is underestimated" I think you mean the POPC underestimates particle size in the HTHH plume?

62. Figure 10: What are the bins for this instrument?

63. Figure 10: This is scattering efficiency at 90 degrees, correct?

64. Figure 10 caption: What is meant by "near 785 nm"? This is all theoretical work, so didn't you define wavelength? Can this be more precise?

65. lines 392-401: There is a lot of redundancy with previous sections. Please condense.

66. lines 396-397: COBALD and SAGE already defined.

67. line 399: "...(see Methods)..." No such section exists and the promised explanation is nowhere to be found in the text. Please correct. Must include detailed description of how you calculate extinction for POPS and COBALD.

68. last paragraph on page 19: I must disagree with your conclusions here. This does not prove sea salt was present. What it shows, is that by changing the refractive index you can get your extinction profile to be within $\pm 50\%$ of the SAGE value. You then state that the quality of the agreement is dependent on both wavelength and composition (this is contradictory to your overall conclusion). Please revise for clarity and provide additional support for your conclusions or consider removing this section entirely.

69. Figure 11: At this point in the text you have not discussed what is meant by "correction". Please consider revising the text for clarity.

70. lines 426-431: It is unclear why this is at this specific location within the paper or why it is in the paper at all. It does not seem to contribute; please revise or remove.

71. line 451: Here you state the expected Cl/Na ratio is 0.8. Earlier you stated it is 1.8 (line 337/338). Please correct.

72. line 462: Which equations are you referring to? Please provide enough information for the reader to understand what these equations do and provide the reference (Zhang et al. 2021 is not in the bibliography).

73. lines 458-481: It is difficult for me to follow your rationale in presenting this information and I have great difficulty at this point in seeing how your overall research fits together. Please revise for clarity.

74. line 472: "ss-X" is introduced and never used. Should this appear in the table?

75. line 477: "The Ca/Cl ratio was measured to be 0.8...(see Table 5)" Table 5 shows it to be 0.88, which is it?

76. Table 5: I am at a total loss for how to interpret/compare these tables. The units aren't even the same. Even assuming a density of 1 g/mL, the numbers between the top tables and the bottom are vastly different. As presented, this provides far more support for rejecting the hypothesis that sea water was present than it does for accepting the hypothesis. If there is a different interpretation the authors need to provide it.

77. line 541: "...significant deviations between our BraVo aerosol samples, collected at stratospheric altitude...and established oceanic ratios reported by *multiple studies*" (emphasis added). If the authors insist on concluding that sea water was present, then they must provide a robust argument to explain this discrepancy. The burden is on them to provide more than possibilities.

78. line 541: Did the authors collect "blank" samples on the flights? If not, then you cannot discount the possibility of contamination in flight, after landing, in transit to the lab, or during analysis.

79. line 551: "The presence of sea salt in the stratospheric plume is further supported by our analysis of POPC measurements."

    (a) First, the authors claim this is "further support", which indicates they already provided support for stratospheric sea salt, which they did not. On line 541 they stated that their measurements had "significant deviations" from established observations. This explained away by an untested and unproven hypothesis that these deviations might be caused by "fractionation" as if the assertion stands on its own.

    (b) Second, the POPC measurements do not provide support for sea salt. I can only assume they refer to the calculated extinctions; this is after they admit their agreement depends on composition and wavelength (the fact that there is a wavelength dependence indicates that your refractive indices are wrong). It is exceedingly simple to adjust the POPC-based extinction calculation to match SAGE to within $\pm 50\%$. I am not insinuating misconduct on the authors, but we know nothing of how they calculated extinction and what the sensitivities of their assumptions are. In short, this is far from convincing support.

80. lines 552-554: "Therefore, the combined evidence from our BraVo samples and *the comparison with established oceanic ratios provides compelling support* for the hypothesis that the [HTHH] eruption injected significant amounts of sea salts directly into the stratosphere." (emphasis added). What? The authors previously stated there were "significant deviations" (line541) and "...our samples consistently diverge from typical seawater composition" (line 543), now they claim "compelling support"? I am at a loss for how to reconcile this. Please clarify.

81. Page 26: Neither GEOS-Chem/MERRA-2 nor ACE-FTS appeared in Section 2. Please include them there.

82. lines 590-592: The authors present a hypothesis for why sulfate is too low in their sample (supposedly it reacted with calcium). However, if you are removing sulfate *and* Ca through this mechanism, wouldn't this result in the Ca/Na ratio being lower than expected? Your Table 6 shows this ratio to be too high, which contradicts your hypothesis.

83. line 592: The authors speculate that gypsum was produced as a loss mechanism for sulfate. Is there any support for this claim other than "sulfate was too low"?

84. Table 7: Do we really need 8-9 decimal places? Please revise to correctly communicate your level of precision.

85. Page 29: The authors introduce another field mission in which nearly every condition surrounding the eruption is different. This is very difficult to follow and I fail to see how this is germane to the original thesis. Is this presented solely to demonstrate the gypsum is formed as part of volcanic eruptions? If so, the authors fail to deliver on this point by stating "These findings imply that gypsum was present in the ground samples." They *imply*, but do not prove.

86. line 634: Here, the authors nail it! This is the far more likely issue at play. Their sampling was flawed, which biased their entire ion analysis. This should be the end of the story.

87. lines 654-655: It is unclear what the authors mean here. "...the aerosol population could have undergone significant changes leading to the observed differences in our in-situ samples." You're comparing your measurements to those made by ACE-FTS (3 days separated). Are you suggesting that it changed more in 3 days than it did in the preceding 8 months?

88. Conclusions: I did not read the conclusions of this paper. There is too much inconsistency and speculation, while ignoring real issues related to their sampling, to give any merit to any conclusions that could be drawn from this.

---

## Author Comment (AC1)

**RESPONSE TO REVIEWER's COMMENT**

Manuscript id: Egusphere-2025-924

**Manuscript Title: Balloon Observations Suggesting Sea Salt Injection into the Stratosphere from Hunga Tonga-Hunga Ha'apai**

**R-1:** The above manuscript describes a detailed analysis of measurement data derived for a quite unusual volcanic eruption. The authors use satellite, lidar and in situ balloon-borne measurement data for this analysis with a focus on optical and chemical plume particle properties. Even if the number of balloon flights is relatively small, the lack of in situ data points of this event make every single data point valuable. Consequently, the manuscript should be published, although I believe it needs some polishing and restructuring beforehand. The text seems rashly written (there are many mistakes) and the structure needs some improvement and streamlining. For instance information about the scientific background is given at several places in the manuscript (and not restricted to the introduction) and also instruments characteristics are described in different sections. Section headlines are not consistent and it is hard to follow the line of arguments, because it is not obvious, what will be discussed next. Additionally, the manuscript is quite lengthy and should be shortened to focus more on the main messages.

Even if this eruption would be the only event of its kind in a century, it is an interesting real-world experiment made by nature. Nevertheless, I miss the information about how frequently such events are expected in the introduction. As an atmospheric researcher I do not know this, but a volcanologist might be able to give an estimate or an upper limit.

**Authors:** We thank the reviewer for this insightful comment and for emphasizing the importance of contextualizing the HTHH eruption within the broader history of submarine volcanic activity. We completely agree that it is crucial to discuss the frequency of such events to determine their long-term impact on atmospheric chemistry and climate.

We propose to add the following lines to the text, directly incorporating the key points from the reviewer's comment:

"Although the HTHH eruption was unprecedented in the satellite era, it is important to note that submarine eruptions form volcanic plumes that frequently reach the stratosphere (Cronin, 1971; Prata et al., 2020) and that large-scale, caldera-forming submarine eruptions are frequent in historical times (e.g., Mastin and Witter 2000). Therefore the HTHH event provides a unique opportunity to understand the long-term impact of such eruptions on the stratosphere and climate."

Additionally, we will include a note that the 1883 Krakatau subaerial eruption also likely indirectly injected significant amounts of seawater into the stratosphere, albeit through a different

mechanism (remobilization by pyroclastic density currents), further highlighting the relevance of water-rich volcanic injections.

**R-1:** **Specific remarks:** p. 1, l. 32: Even if the measurements have an uncertainty, the SAOD was either lower compared to previous eruptions or it was not, but "appears" seems to me the wrong term.

**Authors:** We thank the reviewer for this important clarification. We agree that "appears" introduces unnecessary ambiguity. Based on our analysis and comparison with satellite observations of previous eruptions, the SAOD resulting from the HTHH eruption was indeed lower. We have revised the sentence to reflect this more definitively. It now reads, "The resulting Stratospheric Aerosol Optical Depth was significantly lower than that derived from satellite observations of previous eruptions.

**R-1:** p. 2., l. 77: Analyzing the plume is relevant, but what is the goal in doing this? Providing a better understanding or even a parametrization to the modelers? Understanding satellite data? It is not clear where the manuscript is heading.

**Authors:** We thank the reviewer for pointing out this lack of clarity regarding the overarching goals of our study. We agree that explicitly stating the objectives will significantly strengthen the introduction and provide a clearer direction for the reader. Our primary goals in analyzing the HTHH plume characteristics, using a unique combination of balloon-borne in-situ measurements and satellite observations, are:
1) to provide a detailed characterization of the chemical composition and microphysical properties of the HTHH aerosol plume, particularly the non-sulfate components resulting from the unprecedented seawater injection.
2) to contribute to a better understanding of the radiative impacts of this unique, water-rich, sea-salt-laden volcanic plume.
3) to provide chemical composition and size information data from our measurements for modelers to constrain model simulations of volcanic plume evolution, aerosol processing in water-rich and/or sea-salt bearing environments, and the subsequent climate impacts.

We have revised the text by adding: This study investigates the unique characteristics of the HTHH plume, with a key focus on the unexpected injection of sea salts into the stratosphere, as revealed by novel balloon-borne in-situ measurements of its chemical and microphysical properties. By integrating these findings with SAOD data, we aim to improve understanding of its radiative impacts and provide valuable constraints for modeling the evolution and climate implications of this unprecedented volcanic event.

**R-1:** p. 5., l. 125/126: What about the Sao Paulo measurements? Do they look similar?

**Authors:** We will be investigating the differences between measurements from Bauru and Sao Paulo in more detail in the new manuscript.

**R-1:** Three kilometers difference in the top of the aerosol layer seems to be much, are there any other measurements available which support the zonal averaging theory?

**Authors**: Yes, 3-km seems to be a lot. We could probably use lidar data from Reunion Island to investigate this but this is out of the scope of this paper. Nevertheless, we will make a comparison with Sao Paulo first before further investigations.

**R-1:** Please explain more in detail the argument of no CALIPSO measurements over Brazil due to SAA, for a non-specialist this is not automatically clear.

**Authors:** We will expand our explanation in the manuscript to clarify that the SAA is a region where Earth's magnetic field is weaker, allowing high-energy charged particles to dip closer to the surface. When low orbiting satellites such as CALIPSO enter into this region of higher charged particle density, the increased radiation can interfere with satellite instrumentation, leading to higher noise because of which data is usually discarded.

We will cite the following relevant paper to provide additional context and support, particularly referencing its discussion in Section 5:

- Parker Coye, A., Willitsford, A.: Global probability of a cloud-free line of sight and continuous cloud size from 17 years (2006–2023) of CALIPSO satellite LiDAR data. Journal of Applied Remote Sensing, Vol. 19, Issue 2, 028503 (April 2025). https://doi.org/10.1117/1.JRS.19.028503

**R-1:** p. 11., l. 249: This paragraph should be part of the introduction, but does not belong here.

**Authors:** We thank the reviewer for this suggestion. As requested, the paragraph from page 11, line 249 has been moved to the Introduction section.

**R-1:** p. 12., l. 271: The sentence containing "… both eruptions …" seems to be copied from somewhere as in Fig. 6 more than two eruptions are shown. This does not fit

**Authors:** We thank the reviewer for pointing out this potential confusion. The sentence on page 12, line 271, referring to "both eruptions," specifically relates to the subsequent comparison made within that paragraph between the Stratospheric Aerosol Optical Depth (SAOD) values of the Raikoke and HTHH eruptions, for which we selected consistent latitude ranges. While Figure 6 indeed displays SAOD data for more than two eruptions to provide a broader historical context

from the GloSSAC database, our immediate analysis in that sentence focuses on the direct comparison between Raikoke and HTHH to highlight the relative SAOD values despite their differing $SO_2$ injections. We will revise the section to enhance this clarity and avoid any potential misinterpretation. P.13., L.267.

**R-1:** p. 15., Discussion on level Pos #2: It is not clear to me, why the discussion on these low level data is needed. This holds as well for Fig. 8. For the main topic of this manuscript, it is not needed. The only reason could be to validate the sample data versus ground station data to show that the sampler and the post-flight analysis works well. Otherwise this discussion should be deleted.

**Authors:** We appreciate the reviewer's thoughtful engagement with our manuscript and the opportunity to clarify the role of the Pos #2 data and Figure 8. Below is our detailed response:
The inclusion of the boundary layer sample (Pos #2) and Figure 8 serves two critical supporting roles that strengthen the manuscript's primary focus on the HTHH plume (Pos #4):

**Sampler and Analytical Validation**
The Pos #2 data provide essential validation of our balloon-borne sampling system and post-flight ion chromatography analysis. The observed Cl:Na ratio of 1.5 closely matches that of seawater (~1.8), demonstrating methodological robustness. This validation is crucial for establishing confidence in the novel instrumentation and protocols used to analyze the plume's composition at higher altitudes. Without this reference point, readers unfamiliar with balloon-based sampling might question the reliability of results from the less-characterized free troposphere.

**Contextualizing Regional Influences**
While the HTHH plume is the central focus, Figure 8 highlights regional influences (e.g., South American wildfire smoke) in the boundary layer. Acknowledging these influences offers a more complete atmospheric context for plume transport and evolution during the campaign. This brief discussion avoids overinterpretation of Pos #2 but underscores the dynamic environment through which the plume traveled, indirectly informing interpretations of background variability in Pos #3.

**R-1:** p. 17., Discussion on Fig. 9 and figure caption: The measurements were made 7-8 months after the eruption. Hence one would assume an already well mixed stratospheric aerosol. But the lower row of plots in Fig. 9 shows something different. What is this reason for this, for me unexpected and still large variability? And is this confirmed by, e.g. model results?

**Authors:** We appreciate the reviewer's insightful observation regarding the unexpected variability in the HTHH plume's size distribution and concentration profiles 7-8 months post-eruption, despite the expectation of a more well-mixed stratospheric aerosol layer at this stage. This comment highlights a key finding of our study concerning the prolonged

inhomogeneity of the HTHH plume. The persistent, albeit large, variability observed in the lower panel of Figure 9 can be attributed to several factors unique to the HTHH eruption:

- **Unprecedented Water Vapor Injection:** The HTHH eruption injected an unprecedented amount of water vapor into the stratosphere (Millán et al., 2022; Schoeberl et al., 2022), exceeding typical volcanic eruptions. This massive water vapor injection could have significantly altered aerosol microphysical processes, such as the initial rapid sulfate formation and particle growth, and potentially influenced particle sedimentation rates and subsequent mixing. The *aerosol* evolution in a highly moisturized stratosphere presents a different challenge.
- **Unique Plume Composition (Sea Salt):** Our study, and others (e.g., Khaykin et al., 2022), demonstrate the presence of significant sea salt in the HTHH plume. Unlike purely sulfuric acid aerosols, sea salt particles are less volatile and have different hygroscopic properties and densities. This unique composition could contribute to a slower or different mixing behavior compared to a pure sulfate plume. The gravitational settling of these potentially larger and denser sea salt particles could also lead to vertical sorting and persistent layering, contributing to observed variability.
- **Complex Plume Structure and Altitude:** The HTHH eruption injected material across a wide altitude range, including into the middle and upper stratosphere (reaching ~57 km, Proud et al., 2022), higher than many previous eruptions. While zonal dispersion is relatively fast, vertical mixing and full meridional mixing can take longer, especially for a plume with such a complex initial vertical structure and potential self-lofting effects due to radiative heating. The observed plume might represent a superposition of different layers evolving at slightly different rates.

We have revised the section and added the following discussion in the manuscript to emphasize that the HTHH plume's unique composition and the massive water injection likely contribute to its protracted and complex mixing behavior, leading to the observed variability even several months after the eruption.

"The observed variations in aerosol size distribution and concentration profiles (Figure 9, lower panel), even 7-8 months after the HTHH eruption, highlight a protracted and complex plume evolution compared to typical volcanic events. This persistent inhomogeneity, unexpected for this timescale, can be attributed to several unique characteristics of the HTHH plume. Foremost is the unprecedented water vapor injection, which fundamentally alters the dynamics of aerosol evolution and microphysical processes. Furthermore, the significant presence of less volatile sea salt particles, as identified in our measurements, likely contributes to slower mixing and persistent layering due to their distinct physical properties and gravitational settling. These factors collectively suggest that the HTHH plume exhibited a more complex and prolonged

mixing behavior than previously assumed for stratospheric aerosols several months post-eruption".

**R-1:** p. 18., l. 376: This paragraph is instrument characteristics description and does not belong here but should be moved to an earlier section.

**Authors:** We thank the reviewer for this constructive comment regarding the placement of the instrument characteristics description. We agree that the detailed description of the Particle plus Optical Particle Counter (POPC) and the methodology for deriving extinction profiles (currently in Section 6) is indeed better suited for a dedicated Methods section.

Therefore, the detailed description of the POPC instrument and its data processing, including the Mie theory and correction factors (originally found on page 18, lines 376-388), has been consolidated under a new subsection (**2.2. Balloon-borne Instruments and Data Processing**), within Section **2. Satellite and balloon measurements.** This ensures that all instrument characteristics and data processing methodologies are presented together in the appropriate methods section, thereby improving the manuscript's organization and clarity as per the reviewer's suggestion.

**R-1:** p. 19., last paragraph: The discussion about the extinction profiles and also Fig. 10. As we are talking about a volcanic eruption can you rule out that there has not been any absorbing material in the stratosphere? How would the theoretical response functions look like if you assume e.g. 0.5 or 1% BC? And how would the comparison of the extinction in Fig. 11 be affected due to this?

**Authors:** We thank the reviewer for this insightful and important question, and acknowledge that volcanic eruptions can inject absorbing materials, primarily ash, into the stratosphere, but not typically black carbon aerosols. While the eruption was primarily water-rich, the presence of ash was indeed transiently observed. For instance, Sellitto et al. (2022) identified depolarizing layers using a CALIPSO lidar overpass on January 15th, suggesting the presence of ash and/or ice crystals between 34 and 40 km. However, subsequent overpasses did not confirm the persistence of these aerosols, indicating they may have been scavenged quickly.

Regarding the reviewer's question about the theoretical response functions for absorbing aerosols and their effect on extinction comparisons, we have investigated the theoretical response of our instrument in the presence of absorbing aerosols such as Black Carbon (BC) in a way similar to Figure 10. We can provide a figure (e.g., as supplementary material; see below) showing the scattering efficiency for different types of aerosols (PSL, Ammonium Sulfate, SOA, Black Carbon), and also the scattering efficiency of Ammonium Sulfate, SOA, and Black Carbon relative to PSL. For example, our calculations show that the scattering efficiency of BC particles below 0.4 μm is greater than PSL, while the contrary is true above this threshold. Consequently,

if BC were present, the size threshold correction would have the opposite effect: it would reduce the size threshold for particles d < 0.4 μm and increase it for those with d > 0.4 μm.

While further complex calculations would be required to precisely quantify the effect of a specific percentage (e.g., 0.5% or 1%) of BC on the extinction comparison in Figure 11, we believe the presence of significant black carbon from this volcanic eruption is highly unlikely. Given this, we do not anticipate that a detailed investigation into BC's influence would substantially enhance the core messages of this paper, which focuses on the unique water and sea salt injection.

[Figure]

Figure (left) Scattering efficiency for different types of aerosols (PSL, Ammonium Sulfate, SOA, Black Carbon) and Scattering efficiency of Ammonium sulfate, SOA and Black Carbon relative to PSL.

**R-1:** p. 26., l. 564: "Standard temperature" is 273 K, hence 0°C, 20°C or 25 °C is "normal temperature", right?

**Authors:** We thank the reviewer for this important clarification regarding the definition of "Standard Temperature." We acknowledge that while 0°C (273.15 K) and 1 atm (1013.25 hPa) is a widely recognized definition for STP (e.g., by IUPAC), other conventions exist depending on the field. In atmospheric and environmental sciences, 25°C (298.15 K) and 1013.25 hPa is a commonly used "standard ambient temperature and pressure" or "standard laboratory conditions" for reporting concentrations, particularly in air quality and modeling studies where ambient conditions are often closer to this value.

To avoid any ambiguity and ensure consistency, we have revised the text to explicitly state the temperature and pressure conditions under which the concentrations are reported, rather than using the potentially confusing "STP" acronym with a non-standard temperature. Revised text is included in the manuscript.

**R-1:** p. 25., l. 555, the discussion about the sulfate concentration: The particles are sampled at -60°C and then experience +25°C in the sampler for at least 30 min, according to Fig. 5. How does this influence your sampled sulfate, there must be some evaporation?

**Authors:** We thank the reviewer for highlighting this critical point regarding the potential for sulfate evaporation during the sampling process. We acknowledge that the temperature difference between the sampling environment (-60°C) and the warm-up within the sampler (+25°C over at least 30 minutes, as indicated in Figure 5) could theoretically lead to the volatilization of some semi-volatile sulfate species, such as ammonium sulfate or sulfuric acid, particularly if the particles were small. The residence time at the higher temperature was sufficient. Several factors in our experimental protocols and data interpretation might help address this:

1) As larger particles from HTHH tend to settle (according to previous studies), the larger particles collected on our filters would have a lower surface area to volume ratio, minimizing evaporation.

2) While we did not directly measure the sulfate speciation in this study, and the presence of other cations like $Na^+$, $K^+$, and $Ca^{2+}$ in our samples could suggest that sulfate might have been present in less volatile forms, such as sodium sulfate, potassium sulfate, or calcium sulfate, in addition to ammonium sulfate or sulfuric acid. These metal sulfates have significantly higher decomposition temperatures than ammonium sulfate.

3) The warming occurred under reduced pressure within the sampler during ascent/descent, which could potentially influence the evaporation kinetics. However, the exact impact of this on sulfate evaporation in our specific system requires further investigation, which is beyond the scope of this study but warrants consideration in future work.

We acknowledge that some loss of volatile sulfate species during sampling, transportation, and analysis cannot be entirely ruled out. Quantifying this potential loss would require dedicated laboratory experiments simulating the temperature and pressure conditions experienced during our balloon flights, which we recommend for future studies employing similar sampling techniques.

**R-1:** p. 28., l 615 Tab. 7.: Please delete most of the digits in the numbers, three leading digits should be enough. Please provide the information on the campaign in a new column, as by the upper measure you saved space. What does "Volkilau" mean?

**Authors:** We thank the reviewer for addressing this. We have revised Table 7 to display values with three digits. We have added a new column titled 'Campaign' to Table 7. This column now explicitly indicates the source of the samples, categorizing them as 'Hunga Ground Samples' (from Colombier et al., 2023), 'BraVo' (our study), and 'Kilauea/Hawaii' (referring to samples collected during the balloon campaign, after the eruption of Kilauea Volcano Volkilau, as presented in Vernier et al., 2018).

**R-1:** p. 29., l. 639: What does "sea salt relative to other combustion sources" mean? Sea salt is not from a combustion source. And combustion sources have not been discussed beforehand.

**Authors:** We thank the reviewer for pointing out this confusing and inaccurate statement. The sentence has been revised to clarify the high Cl/K ratio in Hawaii and the expected minimal influence of combustion sources at this remote marine location. The revised sentence now reads: "Finally, the consistently high Cl/K ratio observed in Hawaii suggests a strong influence of marine aerosol, specifically sea salt, at this remote location where contributions from combustion sources, which typically exhibit different Cl/K ratios, are expected to be minimal". P.32., L.634

**R-1:** p. 29., l. 646, The whole paragraph: The difference between the different sulfate measurements is discussed here once again. Please merge all sections with this topic and make it only one section, the reader gets lost otherwise.

**Authors:** We acknowledge the concern raised by the reviewer. We will undertake a thorough revision to consolidate all discussions related to the differences between various sulfate measurements into a single, dedicated section. This restructuring will ensure a more coherent and streamlined presentation of this important topic, significantly improving the manuscript's clarity and readability.

**R-1:** p. 31., l. 689: The "presence of biomass burning" is mentioned for the first time in the conclusions. But this should not be the case, either biomass burning plays a role, then it should show-up already in the former sections, or it should be deleted in the conclusions.

**Authors:** We thank the reviewer for highlighting this important inconsistency. We will either integrate discussion of biomass burning into earlier sections if relevant and supported by data, or remove it from the conclusions to ensure all findings are appropriately introduced.

**R-1:** p. 31., l. 708: The discussion in this paragraph, I again miss "evaporation" as potential reason.

**Authors:** We have revised the paragraph which now reads, "The eight-month time difference between the eruption and our measurements further emphasizes the dynamic evolution of stratospheric aerosols through coagulation, sedimentation, **evaporation**, and chemical reactions".

**R-1**: p. 31., l. 710: You probably mean "sulfate particle mass formation" but not new particle formation.

**Authors:** We thank the reviewer for this precise and important clarification. We agree that "sulfate particle mass formation" is a more accurate description than "sulfate particle formation" in the context of growth through coagulation, as it distinguishes this process from new particle formation (nucleation). The revised text now reads, "The increased water vapor likely led to elevated OH radical concentrations, shortening the $SO_2$ lifetime and potentially enhancing sulfate particle mass formation through coagulation".

**Technical corrections:**
p. 1, l. 46: The sentence reads strange, I assume that a "which" and a comma is missing before "are known …", please check.
**Authors:** We thank the reviewer for this helpful suggestion. We have revised the sentence on page 1, line 46, to improve its grammatical structure and clarity.

The sentence now reads: "Unlike major eruptions with a Volcano Explosivity Index (VEI) greater than or equal to 6 (Table 1), which are known for producing long-lasting sulfate aerosols from $SO_2$ emissions that can cool surface temperatures for years (e.g., Mt. Pinatubo, Parker et al., 1996), a significantly lower amount of $SO_2$ (~0.42 Tg) was detected in the atmosphere from the HTHH eruption."

p. 1, l. 46: Referring to table 1 is misleading, as there is only one major (VEI = 6) volcanic eruption listed. I suggest to delete this reference here.
**Authors:** We thank the reviewer for this helpful feedback. We agree that referring to Table 1 in this context might be misleading, as it lists only one major eruption with a VEI of 6 or greater. We have removed the reference to Table 1 from the sentence on page 1, line 46.

p. 1, l. 48: A paper reference to the statement about the 0.42 Tg SO2 is missing.
**Authors**: We thank the reviewer for pointing out the missing reference for the $SO_2$ emission amount. We have added a citation to the sentence on page 1, line 48. **(Carn et al., 2022)**.

p. 4, l. 102: "Therefore, …" is not correct, the above statement is not the reason, why HTHH did not produce a significantly larger SAOD (this is due to the kind of eruption), but why they look similar in the Fig. 1.
**Authors:** We have revised the sentence on page 4, line 102. We replaced "Therefore," with a phrase that more accurately reflects that the observed similarity is due to the chosen integration

ranges for SAOD. The sentence now reads: "Given that the Hunga plume was injected directly into the stratosphere, while a significant portion of the Raikoke plume remained in the lowermost stratosphere (tropopause to 380°K), the overall SAOD values for both eruptions are similar when integrated over these distinct altitude ranges. This explains why the Hunga eruption does not appear to have produced a significantly larger SAOD than Raikoke in this specific comparison, contrary to previous suggestions (Khaykin et al., 2022)."

p. 5, l. 114: I did not know the city of Bauru, hence I looked at google maps and for me the city seems to be only 250 km west to Sao Paulo. Are you sure about the 400 km? Or is it 400 km distance pointing northwest?

**Authors:** We thank the reviewer for carefully checking the geographical details and for pointing out the inaccuracy in the stated distance between Bauru and São Paulo. We agree that the "400 km" figure was an overestimation.

Based on verification, the approximate road distance between Bauru and São Paulo is indeed closer to 330 km west. We have corrected this detail in the manuscript on page 5, line 114, to ensure accuracy.

p. 5, l. 126: Please start a new paragraph after "(SAA)." as the next sentence is about a different Topic.

**Authors:** We thank the reviewer for this helpful suggestion. We agree that starting a new paragraph after "(SAA)" improves the logical flow and clarity of the text, as the subsequent sentences shift focus from CALIPSO data limitations to the specifics of the BraVo campaign and its instrumentation. We have implemented this change in the manuscript.

p. 6, l. 138: Please start the headline of section 3.1 with "Balloon-born instrument …"

**Authors:** We thank the reviewer for this suggestion. As per a previous constructive comment (regarding the placement of instrument characteristics), the detailed description of the Particle plus Optical Particle Counter (POPC) and related methodologies, previously discussed in Section 3.1, has been consolidated. This led to the creation of a new subsection with the updated title: "**2.2. Balloon-borne Instruments and data processing**."

p. 6, l. 140: A space is missing in "2 kg"

**Authors**: We thank the reviewer for spotting this typographical error; the same has been corrected.

p. 6, l. 145: Any reference paper out on the POPC which could be cited in this paragraph?

**Authors:** We thank the reviewer for highlighting the issue with the print quality of Figure 4. We apologize for the visible pixels and the poor resolution. We will provide a higher-resolution version of Figure 4 to ensure better clarity and print quality in the revised manuscript.

p. 8, Fig. 4: The print quality of this figure is poor (pixels visible), please provide a better version.

**Authors:** We thank the reviewer for highlighting the issue with the print quality of Figure 4. We apologize for the visible pixels and the poor resolution.

We will provide a higher-resolution version of Figure 4 to ensure better clarity and print quality in the revised manuscript.

p. 8, l. 190/193: Please write out the acronyms ECC and CFH when you use them the first time.

**Authors**: We thank the reviewer for pointing this out, the text now reads: **Electrochemical Concentration Cell (ECC) / Cryogenic Frostpoint Hygrometer (CFH)**

p. 9, l. 204: The first paragraph can be deleted as this information was already given on page 7 and is repeated later once again.

**Authors:** We thank the reviewer for carefully pointing out the redundancy in the description of the radio-controlled balloon system. We agree that certain information about the system's purpose and capability to extend flight duration, as presented in the first paragraph of page 9, lines 204-207, overlaps with details already provided on page 7 (lines 174-177). We will streamline the manuscript to avoid this repetition, ensuring that the comprehensive description of the radio-controlled balloon system and its general benefits for extending flight duration is primarily detailed in Section 2.2 ("**Balloon-borne Instruments and data processing**"). The paragraph on page 9, lines 204-211, will be revised to focus specifically on the deployment limitations for this particular campaign (e.g., being limited to a single flight with the enhanced system) and the implications for POPC measurements, without restating information on the system's fundamental operation.

p. 10, l. 237: please exchange ", &" with "and".

**Authors:** Done accordingly.

p. 10, l. 238: is the given ratio "12.8 ng/6ml" correct or is it "12.8 ng/ml"? in the former case a space would be missing.

**Authors:** We thank the reviewer for pointing out the ambiguity in the notation of the detection limit. We confirm that the detection limit refers to 12.8 ng as the total absolute mass of an ion that could be detected when extracted into a 6 ml volume from the filter. The text now reads, "12.8 ng (total mass detectable in the 6 ml extract)".

p. 10, Fig. 5: As you write about the detection range of the POPC in Sec. 3.1.1 in terms of radii, it would be good to add at the legend here "particle diameter", at least that is what I assume.

**Authors:** We thank the reviewer for this helpful suggestion. We will add "particle diameter" to the legend of Figure 5 to explicitly state the unit shown in the figure.

p. 11, l. 242: Please delete one of the two "((".

**Authors:** We will correct the typographical error.

p. 11, l. 242: The sentence "The analysis involved …" is a repetition of the short sentence in line 241.

**Authors:** We agree that the sentence is a repetition and have deleted it.

p. 12, Fig. 6: Please add text to the color scale (what is displayed), otherwise the figure is hard to understand. On the y-axis is SAOD or sSAOD displayed, as stated in the text?

**Authors:** We thank the reviewer for their comment. We will incorporate this change into the final manuscript.

p. 12, l. 272: The "Since" must be deleted, otherwise a half-sentence is missing.

**Authors:** We have deleted "Since".

p. 13, l. 282: I´m not a native speaker, but I believe "non-light" is not correct, what should this be darkness? You mean not light absorbing, please check and rephrase.

**Authors:** We have changed "non-light" to "non-light absorbing".

p. 14, l. 329: Please delete the comma after "emissions".

**Authors:** The comma will be deleted.

p. 16, l. 346: "AMSL" is not explained.

**Authors:** We will ensure that "AMSL" (Above Mean Sea Level) is spelled out when it first appears in the manuscript.

p. 17., l. 363: The reference to Fig. 10 seems to wrong, should be Fig. 11, as Fig. 10 shows something different (as correctly described on page 18).

**Authors:** We thank the reviewer for identifying this error, which is now corrected.

p. 18, l. 382: Please exchange "H.(2022)" with "H., 2022)"

**Authors:** We have corrected the reference on page 18, line 382, by exchanging "H.(2022)" with "H., 2022)"

p. 18, l. 384: The "size of the POPC" hopefully stays constant, what you mean is the "particle size derived by the POPC".

**Authors:** We agree that the phrasing "the size of the POPC" was imprecise and could be misleading, as the instrument's physical size remains constant. Our intention was to refer to the particle sizes derived / measured by the POPC. The revised text reads, "The scattering efficiency for both sea salt and sulfate is systematically lower than PSL and therefore, the **particle size derived by the POPC** within the HTHH plume is underestimated".

p. 19, l. 395: The unit for an hour should be "h" not "hr", please exchange.

**Authors:** We have corrected the unit for an hour from "hr" to "h".

p. 19., l. 396: The name of the instrument "Compact Optical …" should be stated in Sec. 3.1.2, but not anymore in the results section.

**Authors:** We agree that the full name of an instrument, such as the "Compact Optical Backscatter Aerosol Detector (COBALD)," should be introduced and defined in the methods section, not in the results section. The same will be introduced in **Section 2.2 ("Balloon-borne Instruments and data processing")**.

p. 19, l. 399: The reference to the "Methods", what is meant with this, an appendix? Or a former section of the manuscript?

**Authors:** We thank the reviewer for requesting this clarification. We apologize for the ambiguity.

The reference to "Methods" on page 19, line 399, refers to the detailed description of the POPC instrument and its data processing, which is provided in Section 3.1.1, "POPC," within the manuscript.

We will update the text to explicitly refer to this section (e.g., "see Section 3.1.1") in the revised manuscript to ensure clarity.

p. 21, l. 448: The format of the Adachi and Buseck reference, there a two opening parentheses but only one closing parenthesis. Some of the references in this section use the "and" others the "&". Please harmonize in the whole document.

**Authors:** We thank the reviewer for their meticulous review of our references and for pointing out these important formatting inconsistencies. We will correct the specific error in the Adachi and Buseck reference on page 21, line 448, by fixing the parentheses. We will conduct a

thorough review of the entire manuscript to harmonize the use of "and" versus "&" across all references.

p. 22, l. 458: Space missing after "eruption".

**Authors:** We thank the reviewer for identifying this typographical error. We have added the missing space.

p. 22, l. 461: The name of the volcano and its abbreviation was already introduced beforehand, hence use only HTHH here. Same line 600. Later in the manuscript many different versions of the name are used "Hunga" samples or plume, "Hunga Tonga" plume, "HungaTHH" etc., please Harmonize.

**Authors:** We thank the reviewer for this comment regarding the inconsistent naming and abbreviation of the Hunga Tonga-Hunga Ha'apai (HTHH) volcano throughout the manuscript. We agree that maintaining consistent terminology is essential for clarity and professionalism.

We will conduct a thorough review of the entire manuscript to harmonize all mentions of the volcano. After its initial full introduction, we will consistently use only the abbreviation "HTHH." This will apply to the specific instances mentioned and all other variations (e.g., "Hunga" samples, "Hunga Tonga" plume, "HungaTHH") to ensure uniformity throughout the document.

p. 22, Tab. 4: Table caption, should be "Zhang". And in the first row should be "Pos# 4" and not "Pos3 4".

**Authors:** We have corrected the table caption to "Zhang." Additionally, we have corrected the typo in the first row, changing "Pos3 4" to "Pos# 4."

p. 23, l. 493: In the reaction equation the charges on both sides are not equal, but this should be the case, right?

**Authors:** We have corrected the table caption to "Zhang." Additionally, we have corrected the typo in the first row, changing "Pos3 4" to "Pos# 4."

p. 23, l. 509: The "4" in CaSO4 should be subscribed,. Please check the whole manuscript.

**Authors:** We have corrected this on page 23, line 509, and will conduct a thorough review of the entire manuscript to ensure that all chemical formulas are consistently and correctly formatted with appropriate subscripts.

p. 23, l. 511: A space is missing in "5 Tg".

**Authors:** We thank the reviewer for identifying this typographical error. We have added the missing space in "5 Tg".

p. 24, l. 520: Table 5 is of poor print quality, partly to small characters and overcrowded with information. How about extracting the essential information from the literature and include it in the newly made lower part of the table?

**Authors:** We thank the reviewer for their valuable feedback regarding the clarity and presentation of Table 5, and agree that the current version suffers from poor print quality. We appreciate the suggestion and will work to improve its readability.

p. 27, l. 592: The space in "hy drated" must be deleted

**Authors:** We will delete the space.

p. 29, l. 633: Space missing before the "μm".

**Authors:** We thank the reviewer for identifying this typographical error. We have added the missing space before "μm".